# USP7 and USP47 deubiquitinases regulate NLRP3 inflammasome activation

Pablo Palazón-Riquelme[1,†], Jonathan D Worboys[1,†], Jack Green[2], Ana Valera[3], Fatima Martín-Sánchez[4], Carolina Pellegrini[5], David Brough[2] & Gloria López-Castejón[1,*]

## Abstract

**The assembly and activation of the inflammasomes are tightly regulated by post-translational modifications, including ubiquitin. Deubiquitinases (DUBs) counteract the addition of ubiquitin and are essential regulators of immune signalling pathways, including those acting on the inflammasome. How DUBs control the assembly and activation of inflammasomes is unclear. Here, we show that the DUBs USP7 and USP47 regulate inflammasome activation in macrophages. Chemical inhibition of USP7 and USP47 blocks inflammasome formation, independently of transcription, by preventing ASC oligomerisation and speck formation. We also provide evidence that the ubiquitination status of NLRP3 itself is altered by inhibition of USP7 and USP47. Interestingly, we found that the activity of USP7 and USP47 increased in response to inflammasome activators. Using CRISPR/Cas9 in the macrophage cell line THP-1, we show that inflammasome activation is reduced when both USP7 and USP47 are knocked down. Altogether, these data reveal a new post-transcriptional role for USP47 and USP7 in inflammation by regulating inflammasome activation and the release of the pro-inflammatory cytokines IL-1β and IL-18, and implicate dual USP7 and USP47 inhibitors as potential therapeutic agents for inflammatory disease.**

**Keywords** deubiquitination; inflammasome; macrophages; USP47; USP7
**Subject Categories** Immunology; Microbiology, Virology & Host Pathogen Interaction; Post-translational Modifications, Proteolysis & Proteomics

## Introduction

Inflammation is the response of the body to tissue injury or infection that is initiated by innate immune cells such as macrophages. Upon detection of pathogen-associated molecular patterns (PAMPs) or damage-associated molecular patterns (DAMPs) by pattern recognition receptors (PRRs), such as Toll-like receptors (TLRs), signalling cascades are activated, including the NF-κB pathway, that result in the upregulation of pro-inflammatory genes including *interleukin-1β* (IL-1β) [1].

The NF-κB pathway is regulated by post-translational modifications (PTMs) such as ubiquitination, a reversible addition of ubiquitin, the removal of which is mediated by deubiquitinases (DUBs). This pathway is an example where fine regulation of the balance between the addition of ubiquitin by ubiquitin ligases such as TRAF3 or TRAF6 and the removal of ubiquitin by DUBs such as A20 or CYLD is crucial for its correct functioning [2,3]. Disruption of the ubiquitin balance has detrimental consequences for health, and dysregulation of DUBs such as A20 is associated with multiple autoimmune or inflammatory disorders such as rheumatoid arthritis and psoriasis [4,5].

In addition to A20 and CYLD, up to 10 other DUBs have been implicated in the control of the NF-κB pathway, including USP7 (or HAUSP) [6,7]. USP7 was first identified as a viral binding protein that preferentially cleaves K11-, K63- and K48-linked ubiquitin chains [8]. USP7 regulates the levels of p53 and its ubiquitin E3 ligase MDM2 (mouse double minute 2 homolog) by preventing their degradation by the proteasome [9]. USP7 can also stabilise other proteins linked to tumorigenesis such as PTEN [10]. More recently, USP7 was reported to regulate NF-κB transcriptional activity in the nucleus, by increasing NF-κB stability [6]. However, similar to A20 and CYLD, cytosolic USP7 can also act as a negative regulator of the NF-κB pathway by mediating the deubiquitination of NEMO that leads to the retention of NF-κB in the cytosol, thus suppressing its activity [7,11]. These reported

1  Division of Infection, Immunity and Respiratory Medicine, Faculty of Biology, Medicine and Health, Manchester Collaborative Centre of Inflammation Research, Manchester Academic Health Science Centre, Core Technology Facility, School of Biological Sciences, University of Manchester, Manchester, UK
2  Division of Neuroscience and Experimental Psychology, Faculty of Biology, Medicine and Health, Manchester Academic Health Science Centre, School of Biological Sciences, University of Manchester, Manchester, UK
3  Departamento de Biología Celular e Histología, Facultad de Biología, Instituto Murciano de Investigación Biosanitaria (IMIB-Arrixaca), Universidad de Murcia, Murcia, Spain
4  Grupo de Inflamación Molecular, Centro de Investigación Biomédica en Red en el Área Temática de Enfermedades Hepáticas y Digestivas, Hospital Clínico Universitario Virgen de la Arrixaca (IMIB-Arrixaca), Murcia, Spain
5  Department of Clinical and Experimental Medicine, University of Pisa, Pisa, Italy
   *Corresponding author. Tel: +44 1612751690; E-mail: gloria.lopez-castejon@manchester.ac.uk
   †These authors contributed equally to this work

roles suggest that USP7 activity can perform opposing functions, depending on cellular localisation and substrate recognition, although how this is achieved is unclear. USP47, which shares 48.4% similarity with the catalytic site of USP7, is its closest related DUB (Appendix Fig S1). Besides their N-terminal catalytic core [12], they present a similar domain structure with a long C-terminal region containing multiple Ub-like domains (Appendix Fig S1) [13]. Different enzymatic properties of USP7 versus USP47 have been shown [14]. To date, the physiological functions as well as enzymatic properties of USP47 remain unclear. Roles described for USP47 are varied, ranging from contributing to DNA repair by controlling DNA polymerase β levels, to maintaining E-cadherin levels and hence contributing to stable epithelial cell–cell adhesion [15,16]. As of yet, no link between USP47 and the immune system has been described.

Recognition of danger signals by macrophages also leads to the assembly of a molecular complex called the NLRP3 inflammasome. This complex is required to recruit and activate caspase-1 leading to the processing and subsequent release of the cytokines IL-1β and IL-18, which are otherwise stored within the cytoplasm as inactive precursor molecules. NLRP3 inflammasome activation in macrophages is considered as a two-step process. First, a priming step involving TLR and NF-κB activation induces the upregulation of *pro-IL-1β* and other inflammasome components such as *NLRP3*. Second, an activating signal (e.g. ATP, pore-forming toxins or crystals) is sensed by the cytosolic PRR NLRP3 leading to the assembly of the inflammasome complex and the activation of the enzyme caspase-1 [17].

In recent years, it has become clear that the assembly of the inflammasome is regulated by the ubiquitin system [18]. As numerous DUBs control different aspects of NF-κB signalling, it is quite plausible that several DUBs are also involved in inflammasome activation. In particular, deubiquitination has proven essential for inflammasome assembly and currently three DUBs (A20, BRCC3 and USP50) have been directly implicated in this process [19–21]. While A20 dampens both NF-κB and inflammasome activation [20,22], BRCC3 and USP50 positively regulate the inflammasome, independently of NF-κB [19,21]. Considering the contribution of A20 to these two pathways and the relationship between USP7 and NF-κB, we investigated whether USP7, and the closely related USP47, contributed to inflammasome activation. We discovered that inhibition of USP7 and USP47 in macrophages impaired inflammasome formation independently of transcription by preventing ASC speck formation. We also found that inflammasome-activating signals regulate USP7 and USP47 activity and that this occurred independently of the priming signal. Concurrently, we provide evidence that the ubiquitination status of NLRP3 itself is affected by inhibition of USP7 and USP47, suggesting a direct role for these DUBs on NLRP3 activation. Finally, we report that while inflammasome activation was seemingly not affected in either USP7- or USP47-deficient macrophages, it was inhibited in double USP7- and USP47-deficient cells, suggesting that both DUBs may contribute redundantly to inflammasome activation. These data provide new insights into the regulation of the NLRP3 inflammasome and identified USP7 and USP47 as potential therapeutic targets for inflammatory disease.

# Results

## Inhibition of USP7 and USP47 impairs NLRP3 inflammasome activation in macrophages

Given the involvement of USP7 in the NF-κB pathway, we sought to investigate the role of USP7 on inflammasome activation. To test this, we used the inhibitor P22077, which inhibits both USP7 and the closely related DUB USP47 (Appendix Fig S1) [23–26]. Human monocyte-derived macrophages (MDMs) were primed with LPS (4 h) and then incubated with the NLRP3 activator nigericin, in the presence or absence of P22077. We observed that P22077 inhibited IL-1β, IL-18 and caspase-1 cleavage and release, as well as pyroptotic cell death (Fig 1A–D), indicating that USP7 and/or the closely related DUB USP47 are important for inflammasome activation. Inhibition of USP7 and USP47 also impaired inflammasome activation in response to the different, well-characterised NLRP3 activators CPPD crystals and ATP. Indeed, P22077 also blocked the processing and release of IL-1β, IL-18 and caspase-1 induced by CPPD crystals, although no reduction in cell death was observed (Fig 1A–D). This suggested that the inhibitor only blocked inflammasome-dependent cell death, given that a previous report has demonstrated that cell death induced by crystals is inflammasome-independent [27]. Given the great variability in P2X7R responses to ATP within human cells, due to the different P2X7R haplotypes present in the human population [28], we used murine bone marrow-derived macrophages (BMDMs) to assess the effect of P22077 on ATP-mediated inflammasome activation. As expected, ATP-dependent cleavage and release of IL-1β and caspase-1 by ATP were inhibited by P22077 (Fig EV1A and B). A reduction in ATP-induced cell death was observed with the well-characterised NLRP3 inhibitor MCC950 although this was not statistically significant (Fig EV1C) [29].

To further validate a regulatory role for USP7 on inflammasome activation, we treated human MDMs with two other USP7 inhibitors, P5091 (which is structurally related to P22077) and HBX19818 (structurally unrelated to P22077) [23,24,26]. We observed the same inhibitory effect on inflammasome activation, where both inhibitors impaired nigericin-induced IL-1β release and cell death (Appendix Fig S2). Since USP7 and USP47 share considerable similarity of their catalytic site, it is possible that these inhibitors also block USP47, although this has not been directly tested.

In order to further dissect the role of USP7 and USP47 on inflammasome activation, we tested whether the inhibition seen with P22077 was specific to the NLRP3 inflammasome. Firstly, we tested this on the activation of the AIM2 inflammasome, which is important for viral immunity and is activated by double-stranded DNA. As for NLRP3, ASC is also required for AIM2 activation [30–32]. Thus, we incubated LPS-primed human MDMs with liposomes containing Poly (dA:dT), a repetitive double-stranded DNA synthetic sequence that activates the AIM2 inflammasome (Fig EV2). Poly (dA:dT) treatment induced an increase in IL-1β and caspase-1 cleavage and release, which was significantly reduced by P22077 (Fig EV2A and C), yet an inhibition of cell death was not observed (Fig EV2B). However, at the time this paper was under review Gaidt *et al* showed that unlike in the mouse, human AIM2 is dispensable for DNA-mediated inflammasome activation in human myeloid

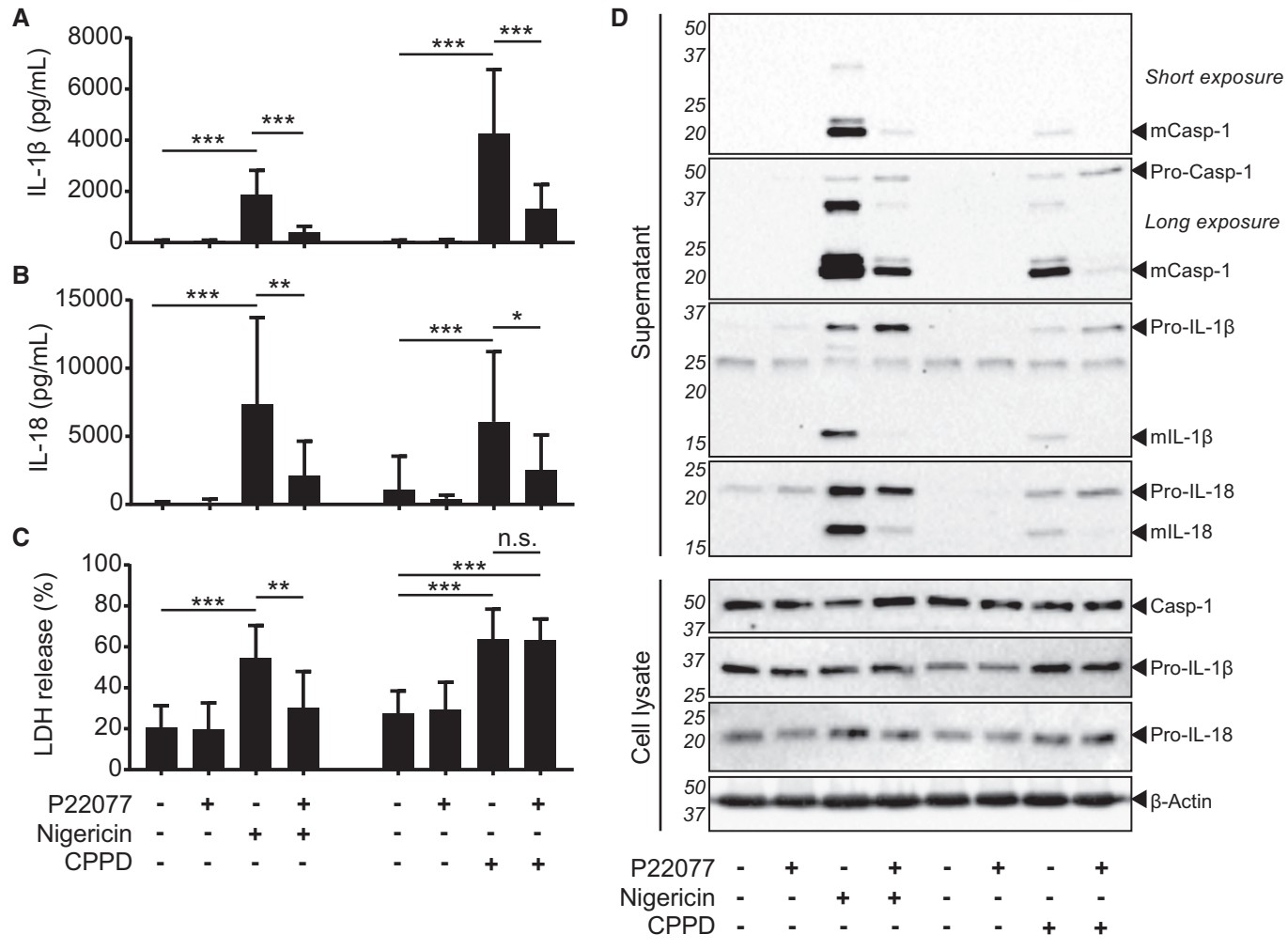

**Figure 1.  Inhibition of USP7 and USP47 with P22077 blocks human NLRP3 inflammasome activation.**

A   IL-1β ELISA of supernatants from LPS-primed (1 μg/ml, 4 h) MDMs pre-incubated with 0.1% DMSO or P22077 (2.5 μM) 15 min before treatment with either nigericin (10 μM, 45 min) or CPPD crystals (250 μg/ml, 2 h). Bars represent the mean ± SD, *n* = 13 and 11 independent blood donors for nigericin and CPPD, respectively. ***P < 0.001 using a one-way ANOVA.

B   IL-18 ELISA of supernatants from MDMs treated as in (A). Bars represent the mean ± SD, *n* = 11 independent blood donors. ***P < 0.001, **P < 0.01 and *P < 0.05 using a one-way ANOVA.

C   LDH release from MDMs treated as in (A). Bars represent the mean percentage of LDH release relative to the total cells lysed ± SD, *n* = 10 and 11 independent blood donors for nigericin and CPPD, respectively. ***P < 0.001; **P < 0.01 using a one-way ANOVA. n.s. = not significant.

D   Western blots of supernatants and cell lysates from MDMs treated as in (A). Bands in the figure represent the following: pro-IL-1β (31 kDa); mature IL-1β (mIL-1β, 17 kDa); pro-IL-18 (24 kDa); mature IL-18 (mIL-18, 18 kDa); pro-caspase-1 (pro-Casp-1, 45 kDa); and mature caspase-1 (mCasp-1, 20 kDa). β-Actin is shown as a loading control. Blots are representative of at least three independent blood donors.

Source data are available online for this figure.

cells. Instead, dsDNA induces a cGAS-STING-mediated lysosomal cell death that leads to NLRP3 inflammasome activation [33]. To confirm this, we performed the Poly (dA:dT) treatment as above in the presence of the NLRP3 inhibitor MCC950. In line with this work, we observed that MCC950 inhibited IL-1β release induced by Poly (dA:dT) but not cell death (Fig EV2D and E), indicating that the inhibitory effect we observed with P22077 was due to NLRP3 and not AIM2 inhibition. Similar results were observed when using THP1 cells (Appendix Fig S3). We then tested the effect of P22077 on AIM2 activation in murine macrophages and observed that a higher dose of P22077 (10 μM rather than 2.5 μM) was needed to

inhibit AIM2 inflammasome activation (Fig EV3A and C). To further support that this inhibition was independent of NLRP3, we tested the effect of P22077 in the presence of the MCC950 inhibitor. Unlike human cells, MCC950 did not block this activation and similar results were observed with P22077 in the presence or absence of MCC950, inhibiting IL-1β release, but having no effect on cell death (Fig EV3B).

We also tested the effect of P22077 on NLRC4 inflammasome activation, which recognises cytosolic bacterial flagellin [34]. P22077 only inhibited flagellin-mediated IL-1β and caspase-1 cleavage and release at a higher concentration, as with Poly (dA:dT) and

no cell death induction was detected after flagellin treatment either in the presence or absence of P22077 (Fig EV3A, B and D). Treatments were also performed in the presence of MCC950 and excluded any implications of NLRP3 activation (Fig EV3A and B).

We finally tested the effect of P22077 on the non-canonical inflammasome. The non-canonical inflammasome is triggered by cytosolic LPS binding and activating caspase-11(in mouse), which leads to the formation of gasdermin D pores in the plasma membrane that in turn activates the NLRP3 inflammasome [35]. For this, we primed BMDMs with Pam3CSK4 (4 h) and then pre-treated with P22077 or MCC950 (15 min) before LPS transfection. Here, we found that P22077 did not affect the ability of cytosolic LPS to trigger IL-1β release, unlike MCC950 that blocked this event, while neither of these inhibitors affected the levels of cell death (Fig EV4). This effect on IL-1β was an unexpected result given that current knowledge states that the underlying mechanisms regulating non-canonical and canonical NLRP3 activation are expected to be the same and depend on $K^+$ efflux. We observed that 10 μM P22077 induced an increase in basal cell death compared to 2.5 μM or vehicle-treated cells. This increase in cell death induced by 10 μM P22077 was not observed in other experiments when incubation times were shorter (4 h, Fig EV3) and was likely due to a toxic effect of the drug after long incubation (24 h) at this concentration. Together, these data demonstrate that USP7 and/or USP47 are involved in the activation of the canonical NLRP3 inflammasome. The lack of effect on AIM2 and NLRC4 inflammasomes at a concentration sufficient to inhibit USP7 and/or USP47 points to a secondary effect of P22077 on other DUBs at higher concentrations.

## Transcriptional inflammasome priming is not essential for USP7- and USP47-mediated inflammasome activation

Given the known regulatory roles of USP7 on the NF-κB pathway, we questioned whether this DUB contributes to the transcriptional changes observed during TLR4-induced inflammasome priming. We treated MDMs with LPS and/or P22077 for 4 h and measured the mRNA levels of IL-1β, IL-18, NLRP3, ASC and caspase-1 by quantitative real-time PCR (Fig 2A). As expected, IL-1β mRNA levels were significantly upregulated after LPS priming, and this was reduced with P22077. Although P22077 seemed to induce a decrease in pro-IL-18 and NLRP3 mRNA levels after LPS treatment, these data were highly variable and no statistically significant changes were detected. No effect of LPS or P22077 was observed in any of the other transcripts measured (Fig 2A). Additionally, Western blot analysis on lysates obtained from these samples revealed that the abundances of the NLRP3 inflammasome-associated proteins did not show obvious levels of reduction with inhibition of USP7 and USP47, with only consistent decreases in the levels of pro-IL-1β (Fig 2B). Similar to mRNA changes, IL-18 protein levels were variable between donors, but overall, we did not observe an inhibition by P22077. Interestingly, we observed a strong increase in the abundance of NLRP3 protein levels after LPS treatment, which was not reflected at the mRNA level and was unaffected by P22077 treatment (Fig 2B).

Priming of the inflammasome is not limited to transcriptional changes, with non-transcriptional priming occurring very rapidly (within minutes) [36,37]. We questioned whether USP7 and/or USP47 were involved in such non-transcriptional inflammasome

priming, as this process is regulated by deubiquitination [37]. Accordingly, MDMs were treated with a short 10-min LPS priming step followed by a pre-treatment with either vehicle or P22077 and a subsequent activation with nigericin. As expected, no IL-1β was detected in cell lysates given that the short TLR stimulation did not allow for IL-1β production. However, since IL-18 is constitutively expressed and does not require NF-κB-induced transcriptional regulation, the processing and release of IL-18 in response to nigericin were detectable and significantly inhibited by P22077 (Fig 2C and D). IL-18 release was accompanied by an increase in cell death, which was also significantly decreased in the presence of P22077 (Fig 2E). These data suggested that although P22077 can affect IL-1β expression, the contribution of USP7 and USP47 to inflammasome activation can also occur independently of transcription.

## Inhibition of USP7 and USP47 impairs ASC speck formation and oligomerisation

NLRP3 inflammasome assembly is closely linked to ASC oligomerisation that leads to the formation of ASC specks in the cell that are required for caspase-1 activation [38]. In order to determine at which level of the NLRP3-activation pathway we were inhibiting the inflammasome, we investigated the effect that P22077 had on both the oligomerisation of ASC and its subsequent speck formation.

To test this, ASC oligomerisation was measured by covalent crosslinking of purified insoluble THP-1 cell lysate fractions. Western blotting of the stabilised ASC-ASC interactions showed that P22077 reduced nigericin-induced ASC oligomerisation in THP-1 cells (Fig 3A). As ASC specks are also pro-inflammatory in the extracellular space [35,36], we analysed the presence of crosslinked ASC oligomers in the supernatants of such treated THP-1 cells and, as expected, P22077 also blocked the release of ASC specks, as shown by the decrease in the extracellular oligomeric forms of ASC (Fig 3B). This correlates with the decrease in pyroptosis after P22077 treatment described above. Correspondingly, P22077 treatment in these cells significantly decreased the number of ASC specks after nigericin treatment, as visualised by immunofluorescence (Fig 3C and D). These data showed that inhibition of USP7 and USP47 prevents DAMP-induced ASC oligomerisation and speck assembly, indicating that these DUBs might be acting upstream of this molecule.

## Inhibition of USP7 and USP47 regulates post-translational NLRP3 modifications

Upon activation, NLRP3 oligomerises to form a platform that recruits ASC inducing its oligomerisation and speck formation [39,40]. We then hypothesised that P22077 could have an effect on NLRP3 oligomerisation and that this would affect its ability to recruit ASC and form a viable ASC speck. To address this, we investigated the effect of P22077 on NLRP3 oligomerisation in response to ATP using blue-native gels as previously reported [41]. BMDMs were primed, pre-treated with P22077 and activated with ATP, as before. Digitonin-solubilised cell lysates were then resolved on native-PAGE and SDS–PAGE. As expected, a marked shift of NLRP3 into an oligomeric complex (~1,200 kDa) was seen after ATP stimulation (Fig 4A). When cells were pre-treated with P22077 before ATP activation, we observed an accumulation of higher molecular

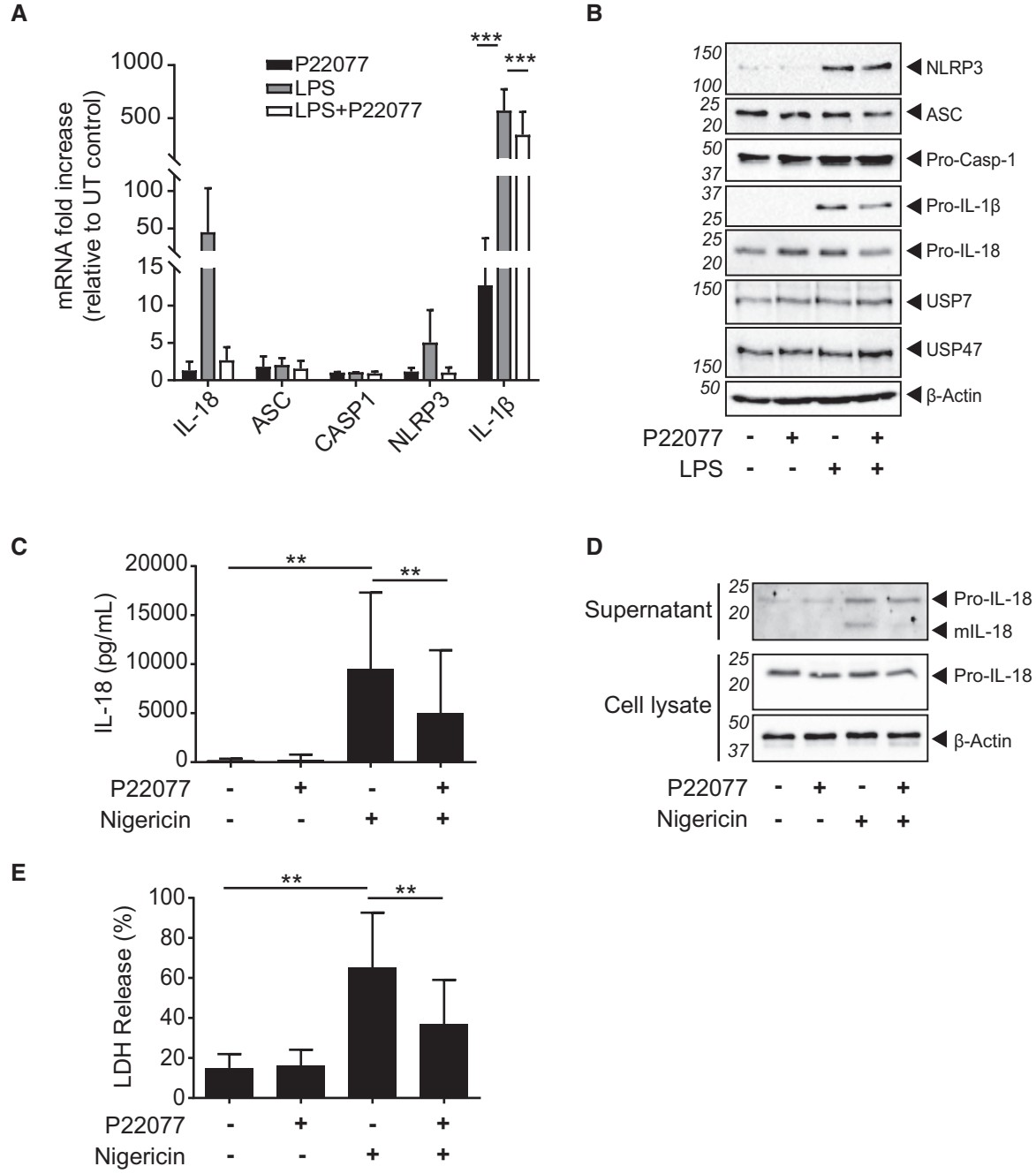

**Figure 2. Transcriptional inflammasome priming is not required for USP7- and USP47-mediated inflammasome activation.**

A   Quantitative RT–PCR showing the mRNA expression levels of IL-1β, IL-18, ASC, Casp-1 and NLRP3 in MDMs treated with P22077 (2.5 μM) and/or LPS (1 μg/ml; 4 h), as indicated. Bars represent the mean fold increase from an untreated (UT) control ± SD, $n = 4$. ***$P < 0.001$ using a one-way ANOVA comparing the mean of each column to the LPS-treated sample.

B   Western blots showing the relative abundance of NLRP3, ASC, pro-caspase-1, pro- IL-1β, pro-IL-18, USP7 and USP47 in cell lysates of MDMs treated with P22077 (2.5 μM) and/or LPS (1 μg/ml) for 4 h, as indicated. β-Actin is shown as a loading control. Blots are representative of at least three independent blood donors.

C   IL-18 ELISA of supernatants from LPS-primed (1 μg/ml, 10 min) MDMs pre-incubated with either vehicle (0.1% DMSO) or P22077 (2.5 μM) for 15 min before treatment with nigericin (10 μM, 45 min). Bars represent the mean ± SD, $n = 9$ independent blood donors. **$P < 0.01$ using a one-way ANOVA.

D   Western blots of supernatants and cell lysates from MDMs as treated in (C). Bands in the figure represent the following: pro-IL-18 (24 kDa); and mature IL-18 (mIL-18, 18 kDa). β-Actin is shown as a loading control. Blots are representative of at least three independent blood donors.

E   LDH release from MDMs as treated in (C). Bars represent the mean percentage of LDH release relative to the total cells lysed ± SD, $n = 9$ independent blood donors. **$P < 0.01$ using a one-way ANOVA.

Source data are available online for this figure.

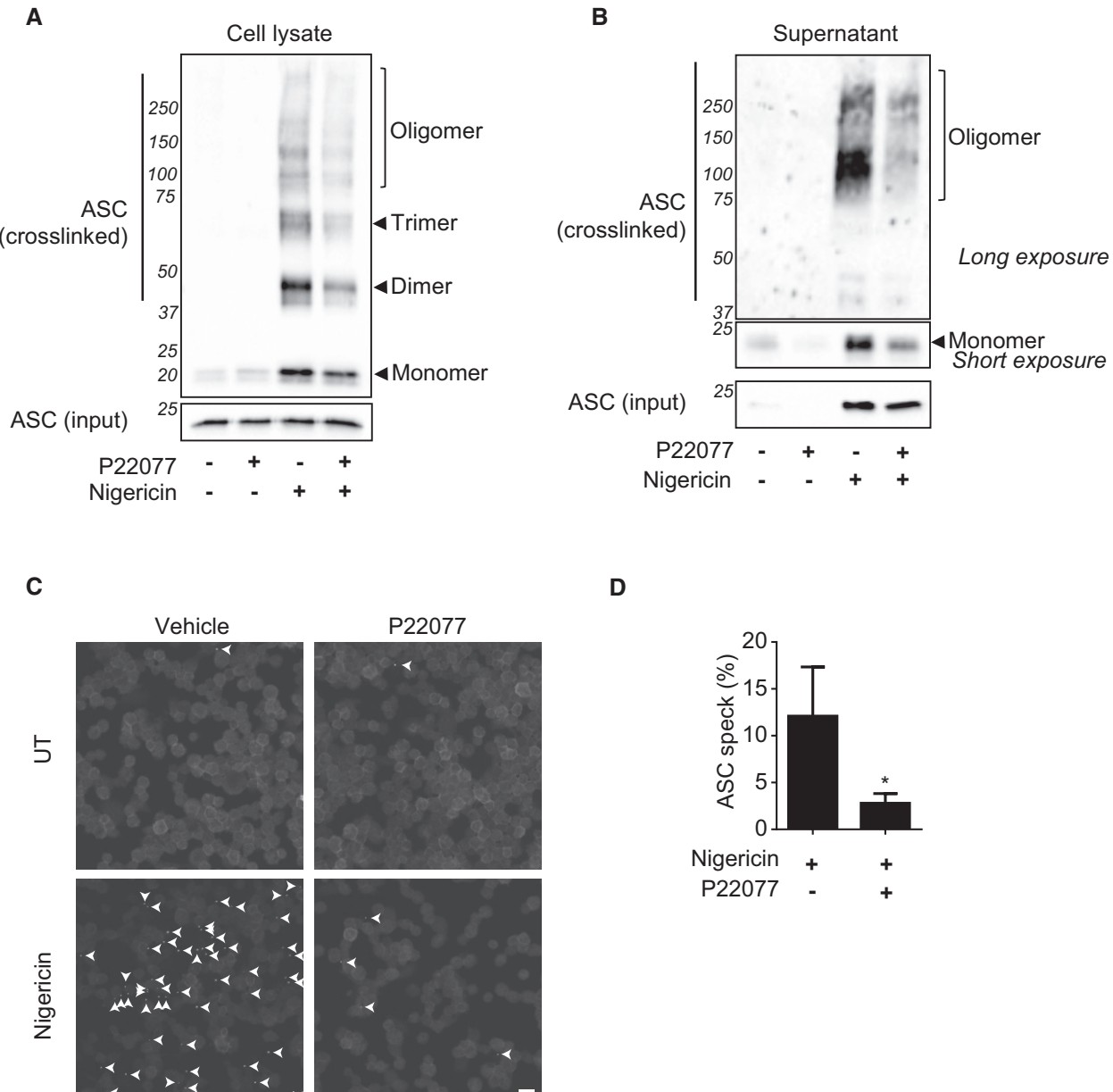

**Figure 3. Inhibition of USP7 and USP47 impairs ASC speck formation and oligomerisation.**

A Western blots showing ASC oligomerisation after DSS-mediated crosslinking in insoluble fractions from THP1 cell lysates. PMA-differentiated THP1 cells were LPS-primed (1 μg/ml, 4 h) and pre-incubated for 15 min with either 0.1% DMSO or P22077 (2.5 μM), followed by treatment with nigericin (10 μM, 45 min). The "input" represents the abundance of ASC in cell lysates prior to isolation of the insoluble fraction. Blots show membranes that were probed with an anti-ASC antibody. Data are representative of at least three independent experiments.

B Western blots showing ASC oligomerisation after DSS-mediated crosslinking in supernatants from THP1s treated as in (A). Blots show membranes that were probed with an anti-ASC antibody. The input represents non-crosslinked supernatant. Data are representative of at least three independent experiments.

C Images from an immunofluorescence experiment showing ASC speck formation (indicated by arrows) in THP1 cells treated as in (A). All images show cells that were stained with an anti-ASC antibody. Images are representative of three independent experiments. Scale bar 20 μm.

D Quantification of ASC speck formation in images depicted in (C). The number of ASC specks was quantified and expressed as the percentage of cells containing specks. Bars represent the mean ± SD, n = 3 independent biological replicates. *P < 0.05 using a t-test analysis.

Source data are available online for this figure.

weight complexes containing NLRP3 (Fig 4A). This was surprising given that, based on the current knowledge [39,40] and the fact that P22077 prevented ASC speck formation, one would assume that

inhibition of USP7 and USP47 would prevent formation of NLRP3-containing oligomers. However, as the increase in higher molecular weight complexes containing NLRP3 was dose-dependent, we

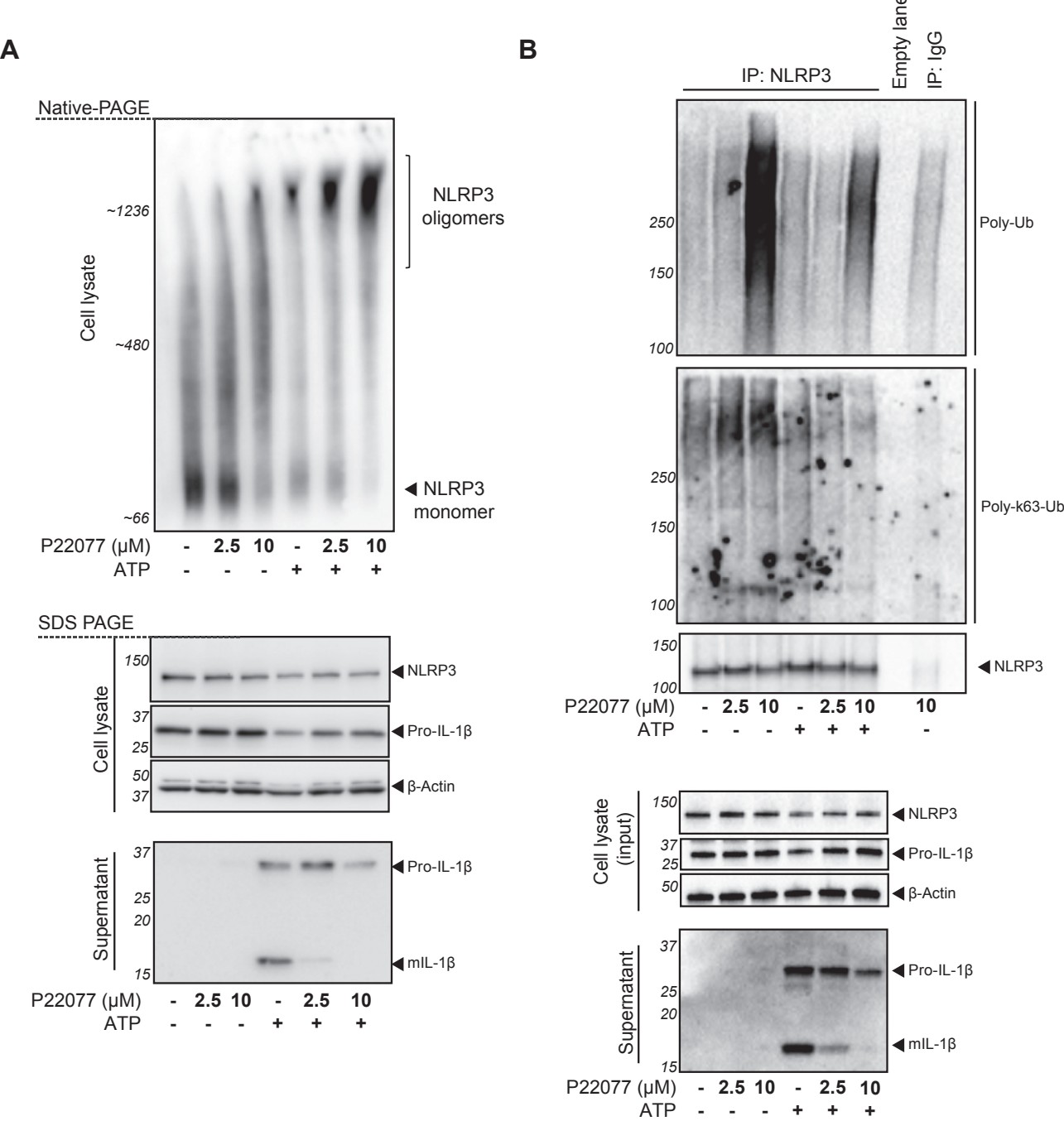

**Figure 4. Inhibition of USP7 and USP47 regulates post-translational NLRP3 modifications.**

A NLRP3 oligomerisation was analysed by native-PAGE. LPS-primed (1 μg/ml, 4 h) murine BMDMs pre-incubated with either 0.1% DMSO or P22077 (at the indicated concentrations) for 15 min before treatment with ATP (5 mM, 30 min). Digitonin-solubilised cell lysates were analysed by native-PAGE or SDS–PAGE and NLRP3 Western blotting. Western blots of supernatants and cell lysates for IL-1β are also shown as a control for P22077 inflammasome inhibition. Bands in the figure represent the following: pro-IL-1β (31 kDa); mature IL-1β (mIL-1β, 17 kDa); and NLRP3 (120 kDa). β-Actin is shown as a loading control. Blots are representative of four independent murine donors.

B NLRP3 ubiquitination was analysed by Western blot. LPS-primed (1 μg/ml, 4 h) murine BMDMs pre-incubated with either 0.1% DMSO or P22077 (at the indicated concentrations) for 15 min before treatment with ATP (5 mM, 30 min). Endogenous NLRP3 was immunoprecipitated (IP) with anti-NLRP3 and probed with either anti-ubiquitin (Poly-Ub), anti-K63-ubiquitin (Poly-K63-Ub) or anti-NLRP3. As a negative control, cell lysate made from LPS-primed and 10 μM P22077-treated cells was immunoprecipitated with a non-specific IgG. Total levels of NLRP3 in cell lysates (input) were also probed. Western blots of supernatants and cell lysates for IL-1β are also shown as a control for P22077 inhibition. Bands in the figure represent the following: pro-IL-1β (31 kDa); mature IL-1β (mIL-1β, 17 kDa); and NLRP3 (120 kDa). β-Actin is shown as a loading control. Blots are representative of two independent murine donors for 2.5 μM P22077 and four independent murine donors for 10 μM.

Source data are available online for this figure.

hypothesised that it could be due to increased ubiquitination of NLRP3 after USP7 and/or USP47 inhibition. Thus, we next investigated the effect of inhibition of USP7 and USP47 on the ubiquitination state of NLRP3. To test this, BMDMs were primed with LPS as before and then activated with ATP for a shorter time (30 min), to reduce the loss of activated cells after pyroptosis. We then immunoprecipitated endogenous NLRP3 with an anti-NLRP3 antibody and analysed its ubiquitination status by Western blotting. An isotype-matched IgG control for the immunoprecipitation was included, showing that the level of ubiquitination detected was above background. NLRP3 is regulated by K48- and K63-linked ubiquitin chains [18], and USP7 and USP47 can cleave K48-, K63- and K11-linked ubiquitin chains [8,14] so we first analysed the effect of P22077 on total ubiquitination. We observed that pre-treatment with P22077 at 10 μM induced a substantial increase in total NLRP3 ubiquitination and that this was slightly reduced after ATP treatment (Fig 4B), suggesting that USP7 and USP47 may regulate NLRP3 receptor ubiquitination. We next analysed NLRP3 ubiquitination status using a K63-linked ubiquitin-selective antibody. We observed that pre-treatment with P22077 induced a small increase in NLRP3 K63 ubiquitination state that disappears after ATP treatment (Fig 4B). This, together with the fact that total NLRP3 protein levels did not change after P22077 treatment, suggests that neither K48- nor K63-linked ubiquitin chains on NLRP3 are regulated by USP7 and/or USP47 after activation of the inflammasome.

## USP7 and USP47 are regulated post-transcriptionally in response to inflammasome-activating signals

Of the DUBs currently known to be involved in inflammasome activation, only A20, a negative regulator of the complex, is transcriptionally upregulated by TLR4 activation [39]. To test whether USP7 and USP47 were regulated in a similar manner, MDMs and THP-1 cells were treated with LPS (4 h) and their mRNA expression was analysed by qPCR, alongside IL-1β (used here as a positive control). As expected, IL-1β mRNA was upregulated in response to LPS. No significant changes in the mRNA levels of USP7 and USP47 were detected in THP-1 cells, while only a twofold increase in USP7 mRNA was detected in MDMs (Appendix Fig S4). However, no changes in USP7 or USP47 protein levels were detected after LPS treatment (Fig 2B). The expression of BRCC3, a DUB also shown to regulate inflammasome activation, was reduced by half in THP1 cells but was unaffected by LPS treatment in MDMs (Appendix Fig S4).

DUB function can also be regulated at the level of its enzymatic activity. Therefore, we assessed whether USP7 and USP47 activity was altered in response to either inflammasome priming or activating signals using activity-based probes (ABPs; [42]). ABPs are suicide substrates that specifically and irreversibly bind to the catalytic site of DUBs in an activity-dependent manner. Active DUBs, which have bound to the probes, can be observed by an approximately 9 kDa increase in their molecular weight by SDS–PAGE. In order to investigate changes in DUB activity, lysates of untreated or LPS-treated MDMs were incubated with the ABP TAMRA-Ub-VME prior to analysis by SDS–PAGE and immunoblotted with antibodies recognising either USP7 or USP47. In control lysates, i.e. those with an absence of ABP, a single major band was observed at the expected molecular weight for USP7 and USP47 (Fig 5A–D; unlabelled). In the lysates that were labelled with ABP, two bands can

be observed (Fig 5A–D; labelled): the lower band corresponding to the inactive non-ABP-bound DUB and a higher band representing the active DUB covalently bound to the ABP probe (Fig 5A–D). LPS treatment did not induce any differences in either USP7 or USP47 activity (Fig 5A and B).

Next, we examined whether activating signals that induce inflammasome assembly affected the activity of USP7 and USP47. Strikingly, treatment of LPS-primed MDMs with nigericin induced a dramatic increase in the proportion of ABP-bound USP7 and USP47 (Fig 5A). The activity changes observed were evident as early as 5 min after nigericin treatment and increased until 45 min of nigericin treatment (Fig 5A), demonstrating a rapid and extensive activation. Nigericin-induced activation of both USP7 and USP47 was not dependent on LPS priming (Fig 5A, final lane). Although nigericin did not affect total levels of USP7 (Fig 5A), an increase in total USP47 levels following nigericin treatment was occasionally observed (Fig 5A). Thus, for USP47, increased enzyme expression may contribute to the increase in overall activity induced by nigericin. Additionally, we observed a slight shift in the migration of USP47 after nigericin treatment, suggesting that this protein may be post-translationally modified (Fig 5A). To see whether activity modulation of USP7 and USP47 was specific to nigericin, this experiment was repeated with CPPD crystals, and similar results were obtained (Fig 5B).

Given the inhibitory effect of P22077 on inflammasome activation, we tested whether P22077 would impair the activity increase in USP7 and USP47. We observed a dose-dependent inhibition of labelling with both DUBs (Fig 5C). At the concentration used in this experiment, we observed that P22077 did not cause a complete inhibition of the nigericin-induced increase in activity of USP7 and USP47 that may be due to the irreversible action of the ABP versus the reversible inhibitor [24].

To better understand how these activating signals contribute to USP7 and USP47 activation, we evaluated whether high extracellular potassium, a well-known inhibitor of the NLRP3 inflammasome, affected these activity changes. Despite the blockade of IL-1β cleavage and release, no inhibition of USP7 or USP47 activity changes induced by nigericin was observed (Fig 5D).

Overall, these data show that danger signals involved in inflammasome activation regulate USP7 and USP47 at a post-transcriptional level in human macrophages demonstrating a new and strikingly important level of control of these DUBs during the inflammatory process.

## Both USP7 and USP47 are required for inflammasome activation

In order to differentiate the involvement of USP7 and USP47 in inflammasome activation, we generated both single and double USP7- and USP47-deficient THP-1 cells using CRISPR/Cas9. As USP7 deficiency had been reported to be lethal in several cases [43], we used a doxycycline-inducible CRISPR system [44] and observed notable reductions in the expression of both proteins with at least two different guide RNAs (Appendix Fig S5). Each of these cell lines responded normally in the absence of doxycycline to nigericin treatment, with no differences detected in the processing and release of mIL-1β and mCasp-1 (Fig 6A). We found that treatment with doxycycline alone induced a slight inhibition of inflammasome activation in our control cell line (those expressing Cas9 alone), probably due to the anti-inflammatory effects of this antibiotic [45]. Despite this,

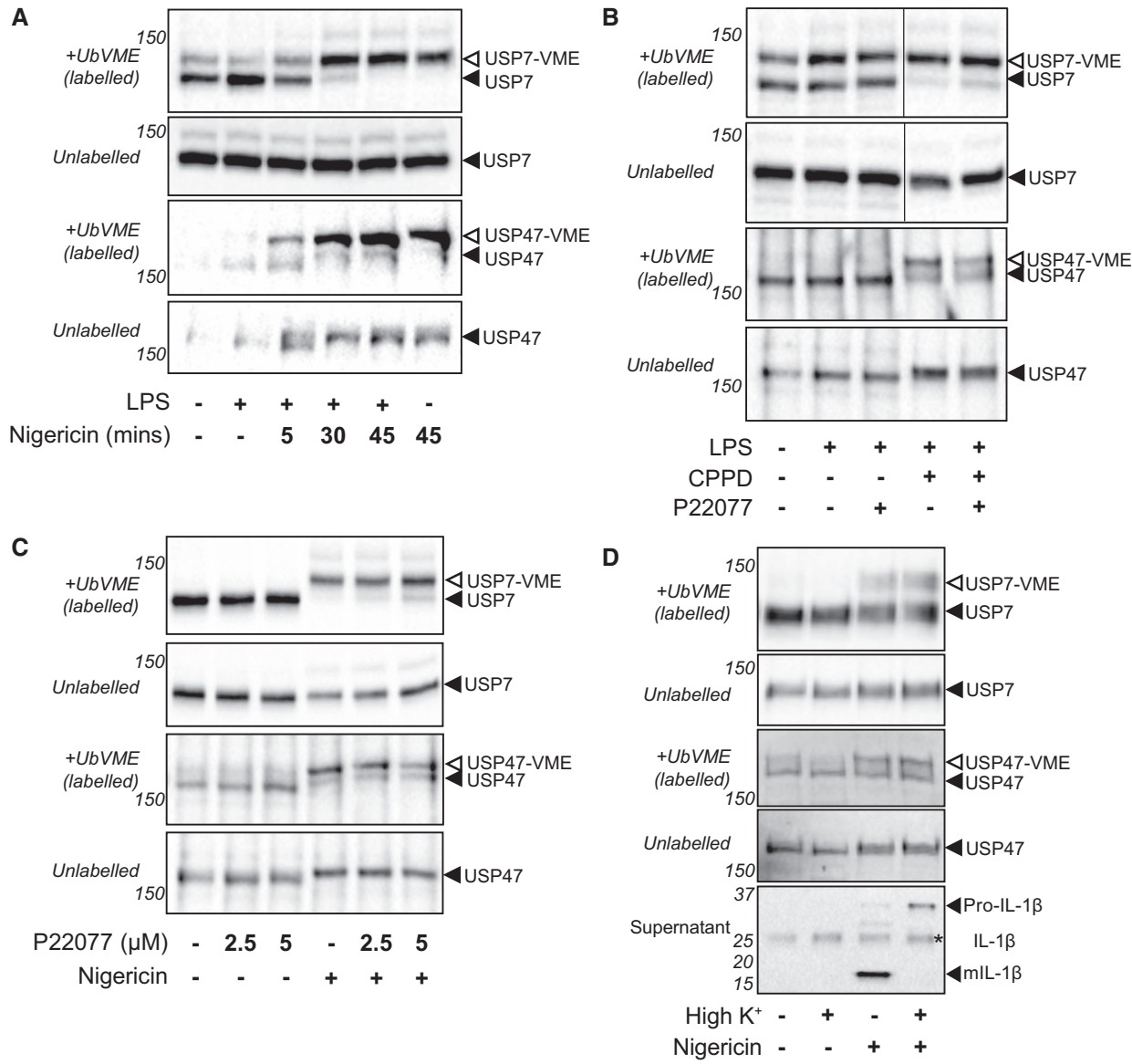

**Figure 5. USP7 and USP47 are regulated post-transcriptionally in response to danger signals.**

A   Western blots showing activity changes in both USP7 and USP47 upon treatment with nigericin. MDMs were either unprimed or LPS-primed (1 µg/ml, 4 h) and then treated with nigericin (10 µM) for differing lengths of time, as indicated. Relative, activity-based labelling of USP7 and USP47 was analysed by Western blot after incubation with ABPs. Higher panels show lysates incubated with ABPs (labelled), with the active DUB bound to ABP indicated by the suffix "–VME". Lower panels, without probe labelling, show total USP7 or USP47 levels (unlabelled). Western blots are representative of three independent MDM donors.

B   Western blots showing activity changes in both USP7 and USP47 upon treatment with CPPD. MDMs were either unprimed or LPS-primed (1 µg/ml, 4 h) and then pre-incubated with either 0.1% DMSO or P22077 (2.5 µM) prior to incubation with CPPD crystals (250 µg/ml, 3 h). Blots are labelled as in (B) and are representative of three independent MDM donors.

C   Western blots showing activity changes in both USP7 and USP47 in LPS-primed (1 µg/ml, 4 h) MDMs pre-incubated with different P22077 concentrations, as indicated, prior to activation by nigericin treatment (10 µM; 45 min). Blots are labelled as in (B) and are representative of three independent MDM donors.

D   Western blots showing activity changes in both USP7 and USP47 upon activation in high extracellular potassium (High K$^+$) conditions. MDMs were LPS-primed (1 µg/ml, 4 h) and then changed to media that either had normal or high concentrations of K$^+$, and subsequently treated with nigericin (10 µM; 45 min). Blots are labelled as in (B) and are representative of at least two independent MDM donors. Pro-IL-1β (31 kDa) and mature IL-1β (mIL-1β, 17 kDa) in the supernatants of these cells are shown as a control. Non-specific bands are highlighted with an asterisk.

Source data are available online for this figure.

we observed an obvious inhibition of the release of both pro and mature forms of IL-1β and caspase-1 release in USP7- and USP47-deficient cells by Western blotting, while the absence of either USP7 or USP47 alone had no visible effect (Fig 6A). In order to quantify this, we also measured mIL-1β release by ELISA and expressed the data relative to its non-doxycycline-treated control, to account for

the effects of the antibiotic. In agreement with the Western blot data, we found that inducing USP7 deficiency had no effect on the levels of mIL-1β release or cell death induced by nigericin (Fig 6B and C). However, inducing USP47 deficiency led to a significant decrease in mIL-1β release induced by nigericin (69% of non-doxy-cycline control in Cas9 alone cells versus 32% in USP47-deficient cells; Fig 6B) and a slight (albeit not significant; 95% versus 76%; Fig 6C) decrease in cell death. Induced deficiency of both DUBs showed impaired mIL-1β release to a similar level as USP47 deficiency (27%), but resulted in a greater, significant reduction in cell death (52%). Similar results were obtained when DUB deficiency was achieved using a different set of RNA guides (Appendix Fig S5), although the effects of USP47 alone were less pronounced in this case. This could be explained by the fact that this USP47 guide resulted in less of a knock-down than with the first (compare Fig EV6A and B).

Similarly, we observed that inducing cGAS-STING-mediated NLRP3 activation with Poly (dA:dT) as described above also depended on USP7 and USP47 [33]. Using Western blotting, a clear reduction in both mature IL-1β and active caspase-1 release in USP7- and USP47-deficient cells was observed (Fig EV5A). Quan-tification of mIL-1β in the supernatants of these cells by ELISAs showed significant reductions in both individually and doubly defi-cient cells (Fig EV5B). In agreement with our P22077 inhibitor data, no differences in the levels of cell death that is mediated by STING [33] and no caspase-1 were detected (Fig EV5C).

To test that this inhibitory effect was not due to an alteration in the priming step, we analysed the levels of expression of pro-IL-1β in the lysates of these cells (Figs 6A and EV5A) and found that there were no apparent reductions in total levels following LPS treatment. We also measured levels of TNF-α and IL-6 (cytokines that are regu-lated by TLR activation) in the supernatants of the CRISPR/Cas9-edited cells upon treatment with LPS (4 h) and observed no dif-ferences in the levels secreted by the different cell lines after doxycy-cline treatment (Appendix Fig S6). In addition to the previous results presented in this study, these data confirm that USP47 and USP7 contribute to NLRP3 inflammasome activation in a non-tran-scriptional manner and provide evidence for a functional redun-dancy between these two DUBs in inflammasome regulation.

## Discussion

The role of ubiquitin as an important regulator of the inflammasome has become more widely appreciated in recent years, with DUBs being key contributors to this process [46]. However, we are yet to fully understand which DUBs control this inflammatory event and how they do so. This study shows that USP7 and USP47 are regu-lated at the activity level by inflammasome-activating signals and that these DUBs are mediators of the NLRP3 inflammasome activa-tion, providing a new mechanistic and molecular insight.

Although our data clearly showed that inhibition of USP7 and USP47 impaired NLRP3-mediated responses, we observed that this inhibition did not affect the activation of the NLRP3 non-canonical inflammasome. This resembles the ubiquitin ligase Pellino2, a newly described mediator of the NLRP3 inflammasome that equally does not contribute to the non-canonical inflammasome [36]. The mechanisms by which the non-canonical inflammasome is regulated

are not well understood, and these data suggest that there might be important differences in the contribution of the ubiquitin system to this process.

The involvement of the ubiquitin system in other inflammasomes such as AIM2 or NLRC4 is also poorly described. There is evidence suggesting that NLRC4 is regulated by ubiquitin [47], but no endogenous ubiquitination of this receptor has yet been described. We found that P22077 inhibition impaired AIM2 and NLRC4 activa-tion in murine macrophages but only at the higher concentration of 10 μM. This could be due to the fact that a different protein is involved in these inflammasome activations or that at higher concentrations the drug might be targeting other DUBs affecting AIM2 and NLRC4. These data however add to the evidence that AIM2 and NLRC4 can indeed be regulated by the ubiquitin system and constitute an interesting new field of research.

NLRP3 oligomerisation into higher molecular weight multi-protein complexes is a key step for inflammasome formation [41]. NLRP3 oligomers are essential to recruit ASC and induce its assem-bly into oligomers and specks, an event crucial for a functional inflammasome [39,40] We saw that inhibition of USP7 and USP47 impaired ASC oligomerisation and ASC speck formation, suggesting that NLRP3 oligomerisation could be affected by these DUBs. Our data showed that inhibition of USP7 and USP47 leads to an increase in oligomeric complexes containing NLRP3. Successful NLRP3 oligomerisation and consequent activation require NEK7 [41]. However, the processes that regulate NLRP3 oligomerisation or the stoichiometry of receptors required to form an active NLRP3 oligomer are unclear. The changes in the molecular weight of the oligomeric complexes containing NLRP3 detected by native gels could reflect the accumulation of aberrant NLRP3-oligomers, the accumulation of highly ubiquitinated NLRP3 forms or both. Whether increased NLRP3 ubiquitination leads to aberrant NLRP3 oligomers that impede the function of NLRP3 as a seed to recruit other inflammasome components is still unknown.

Inhibition of DUB activity with both the broad-spectrum DUB inhibitors G5 and PR960, and the deficiency of BRCC3, leads to an increased ubiquitination of NLRP3 [19,37]. We observed a clear increase in total NLRP3 ubiquitination after P22077 treatment at 10 μM but not at 2.5 μM, while a small increase in K63-linked ubiq-uitination was detected at both 2.5 and 10 μM. After ATP treatment, however, this increase in total ubiquitination was slightly decreased and reversed to basal levels for K63-linked ubiquitination, suggest-ing that upon activation other DUBs such as BRCC3 could be responsible for K63 deubiquitination of NLRP3. USP7 and USP47 preferentially cleave K11-, K48- and K63-linked ubiquitin chains [8,14]. We did not observe changes in total NLRP3 protein levels after P22077 treatment, indicating that it is unlikely that K48-linked proteasomal control is involved. It is then possible that this process is regulated by K11-linked ubiquitin chains, yet it is also possible that USP7 and USP47 do not directly target NLRP3 and that an addi-tional regulated protein, upstream of this process, is mediating this effect. Further work will be required to pinpoint the exact type of chain and mechanism involved in this process.

We found that unlike A20 [22], USP7 and USP47 mRNA levels were not regulated by TLR4 activation in human macrophages. LPS induced no changes in USP7 at the protein level either; however, we observed that USP47 protein levels varied significantly amongst dif-ferent donors. It has been suggested that USP47 levels are regulated

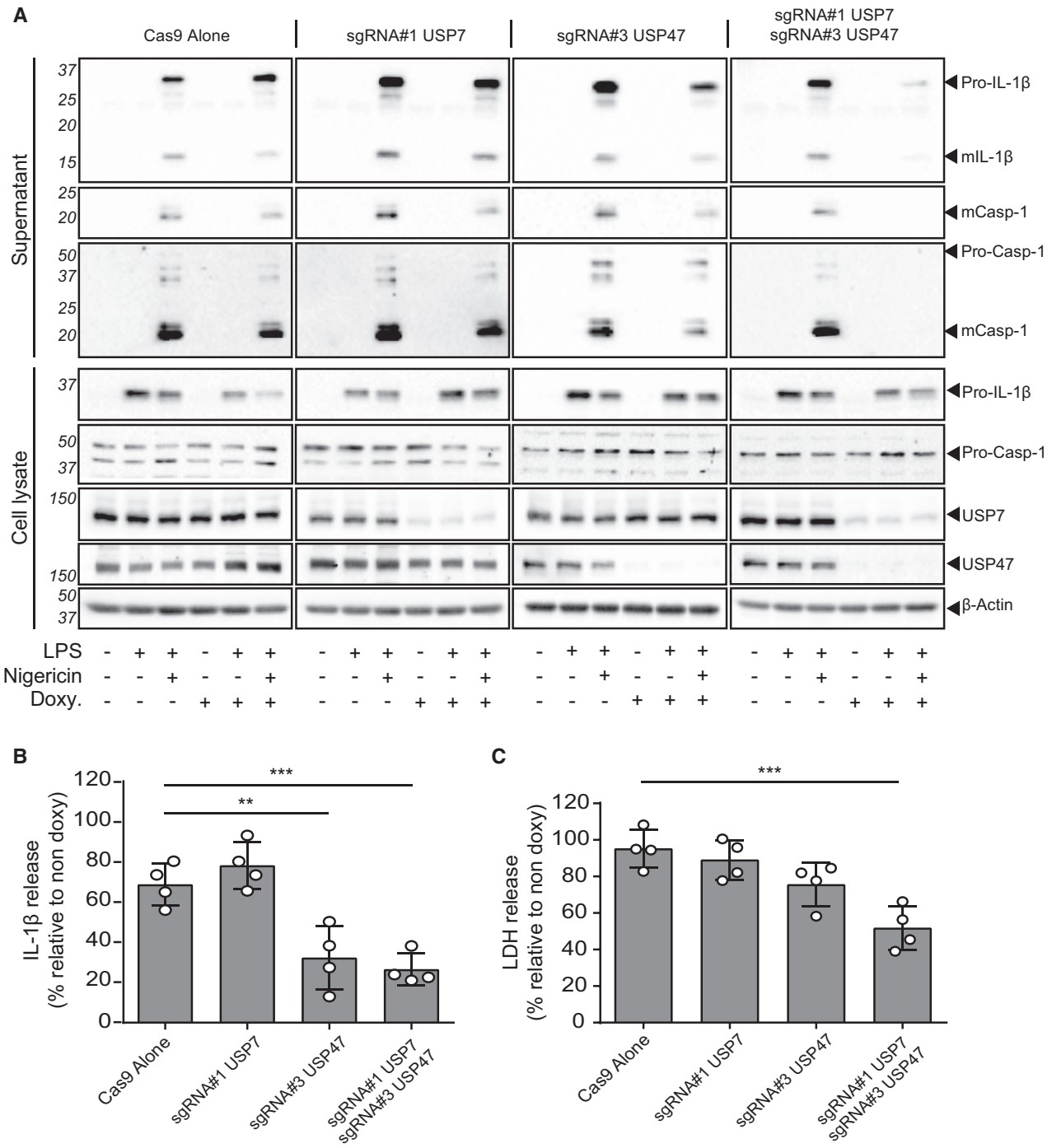

**Figure 6.  Inducible CRISPR/Cas9 knockout for USP7 and USP47 confirms their role in NLRP3 inflammasome activation.**

A    Western blots of supernatants and cell lysates from THP-1 cells engineered by CRISPR/Cas9 to induce USP7 and/or USP47 KOs. Deficiency of both USP7 and USP47 was induced by doxycycline (Doxy) treatment (1 μg/ml; 3 days), as indicated. All cells were differentiated with PMA and either unprimed or LPS-primed (1 μg/ml, 4 h) and treated with nigericin (10 μM, 45 min), as indicated. Bands in the figure represent the following: pro-IL-1β (31 kDa); mature IL-1β (mIL-1β, 17 kDa); pro-caspase-1 (pro-Casp-1, 45 kDa); mature caspase-1 (mCasp-1, 20 kDa); and USP7 and USP47. β-Actin is shown as a loading control. Blots are representative of at least three independent experiments.

B    IL-1β release measured from doxycycline-treated (1 μg/ml; 3 days) PMA-differentiated, LPS-primed (1 μg/ml, 4 h) and nigericin-activated (10 μM, 45 min) THP-1 cells, which contain the indicated sgRNA. Bars represent the mean percentage of IL-1β release relative to their respective non-doxycycline-treated control cells ± SD, $n = 4$ independent biological replicates, plotted as open circles on each bar. **$P < 0.01$ and ***$P < 0.001$ using a one-way ANOVA.

C    LDH release measured from cells as treated in (B). Bars represent the mean percentage of LDH release relative to their respective non-doxycycline-treated control cells ± SD, $n = 4$ independent biological replicates, plotted as open circles on each bar. ***$P < 0.001$ using a one-way ANOVA.

Source data are available online for this figure.

by the β-TrCP-mediated ubiquitination/proteasome pathway [48], so these differences could be explained by variations in proteasome activity in macrophages from different donors due to age or gender [49].

Importantly, we showed that inflammasome-activating signals induced a rapid and sustained increase in the activity of USP7 and USP47. This constitutes the first report placing pro-inflammatory triggers PAMPs and DAMPs as modulators of deubiquitinase activity. Inflammasome-activating signals, such as the ones used here, induce multiple changes to macrophages including ROS production, potassium efflux and lysosomal destabilisation, all of which result in caspase-1 activation and the creation of membrane pores by gasdermin-D eventually leading to pyroptosis [50]. ROS production has been shown to impair DUB activity [51], suggesting this might not be the mechanism driving USP7 and USP47 activity increases. It has been reported that changes in pH or divalent cations can affect USP7 activity [52]. Little is known about how ion changes within the cell affect DUB activity, suggesting a possible regulatory role here. We found that blocking $K^+$ efflux, although sufficient to block inflammasome activity, was unable to prevent both USP7 and USP47 activation, showing that the activity regulation of these DUBs is uncoupled to that of the inflammasome. While $K^+$ ions may not contribute to regulating USP7 and USP47 activity, other ions may still contribute to these changes. We often found that inflammasome activators increased USP47 protein levels, but not USP7, which could be an important way to regulate its function. DUBs are regulated by post-translational modifications [53], and in particular, USP7 can be regulated by phosphorylation and ubiquitination [54]. Little is known about USP47, but the changes observed in its gel migration after nigericin stimulation suggests that they could be regulated by PTMs, in this context, and might be responsible for the subsequent DUB activity increases and/or increased protein levels we describe.

USP47 and USP7 have common structural features and significant similarity in their USP domains, and our results suggest that they behave redundantly in regulating inflammasome activation. We only observed a reduction in IL-1β cleavage and release in both ELISAs and Western blots when both DUBs were absent, and observed a similar phenotype with regard to caspase-1 activation and pyroptosis. This is not the first time that functional redundancy has been suggested to exist between USP7 and USP47, and has already been implicated in the Wnt signalling pathway [55]. Further studies will be needed to examine the exact targets of deubiquitination for these DUBs in this setting, to be able to assess such redundancy.

In conclusion, we have identified that USP47 and USP7 are involved in inflammasome activation in macrophages, describing a role for USP47 in the inflammatory response for the first time. Although our data suggest that these DUBs could be playing a direct role in NLRP3 deubiquitination, we cannot rule out the possibility that USP47 and USP7 act on an upstream mediator of inflammasome activation and the precise mechanism of their contribution to inflammasome activation will need further study. There is already great interest in the development of USP7 inhibitors as therapeutic agents, especially in the field of oncology [56,57], and very selective USP7 inhibitors have been recently described [58,59]. Given the new role of USP47 unveiled here, together with the known role for USP7 in immune pathways, dual USP7 and USP47 inhibitors could also become important for the treatment of inflammatory conditions

in the future and, as our data suggest, would be fundamental in targeting inflammasome-related pathologies.

# Materials and Methods

## Antibodies and reagents

LPS (*Escherichia coli* 026:B6); nigericin (N7143); protease inhibitor cocktail (P8340); *N*-ethylmaleimide (NEM, 04259); phorbol 12-myristate 13-acetate (PMA, P8139); Poly (deoxyadenylic-thymidylic) acid sodium salt (Poly (dA:dT), P0833), adenosine 5′-triphosphate disodium salt hydrate (ATP, A2383) and penicillin–streptomycin (Pen/Strep, P4333) were purchased from Sigma. Foetal bovine serum (FBS) was obtained from Gibco. Lipofectamine 2000 (11668027) and Lipofectamine 3000 (L3000-008) and IgG-Dyna-beads (10003D) were from Thermo Fisher Scientific. P22077 (SI9699) was purchased from Life Sensors, HBX18191 from Generon (HY-17540), P5091 from Sigma (SML0770) and MCC950 from Sigma (PZ0280). Calcium pyrophosphate dihydrate (CPPD, tlrl-CPPD), flagellin (tlrl-stfla) and Pam3CSK4 (tlrl-pms) were obtained from Invivogen.

Primary antibodies used for Western blot analysis were anti-human IL-1β (0.1 μg/ml, goat polyclonal, R&D Systems, AF-201-NA), anti-mouse IL-1β (0.1 μg/ml, goat polyclonal, R&D Systems, AF-401-NA), anti-ASC (1 μg/ml, rabbit polyclonal, Santa Cruz Biotechnology, sc-22514-R), anti-human caspase-1 p20 (1:500, mouse monoclonal, Cell Signaling Technology, 3866), anti-mouse caspase-1 (p10 + p12 rabbit monoclonal, Abcam, EPR16883, ab179515), anti-USP7 (0.5 μg/ml, rabbit polyclonal, Bethyl Laboratories, A300-033A), anti-USP47 (0.2 μg/ml, rabbit polyclonal, Bethyl Laboratories, A301-048A), anti-IL18 (0.5 μg/ml, rabbit polyclonal, LifeSpan BioSciences, LS-C313397), anti-NLRP3 (1 μg/ml, mouse monoclonal, Adipogen, AG-20B-0014), IgG2b (Biolegend, 401201), anti-Ub-HRP (1 μg/ml, Mono- and polyubiquitinylated conjugates, FK2, Enzo Life Sciences, BML-PW0150) and anti-β-actin-HRP (0.2 μg/ml, mouse monoclonal, Sigma, A3854). Secondary antibody HRP conjugates used for Western blotting were anti-rabbit-HRP (0.25 μg/ml, goat polyclonal, Dako, P0448), anti-mouse-HRP (1.3 μg/ml, rabbit polyclonal, Dako, P0260) and anti-goat-HRP (0.13 μg/ml, rabbit polyclonal, Sigma, A5420).

## Cells and treatments

THP-1 cells were maintained in RPMI with 10% FBS and Pen/Strep (100 U/ml). Cells were plated at a concentration of $1 \times 10^6$ cells/ml and incubated for 3 h with 0.5 μM PMA. After 3 h, media were removed and fresh media were added. Experiments were carried out the following day. Peripheral blood mononuclear cells (PBMCs) were obtained from the National Blood Transfusion Service (Manchester, UK). Isolation of peripheral blood mononuclear cells (PBMCs) from leucocyte cones from healthy donors was performed by density centrifugation using a Ficoll gradient. The PBMC layer was carefully removed and washed to remove platelets. Monocytes were obtained by incubation of the PBMC layer with magnetic CD14 MicroBeads (Miltenyi, 130-050-201) for 15 min at 4°C and positive selection using a LS column (Miltenyi, 130-042-401). Monocytes were plated for 6 days (at a concentration of $5 \times 10^5$ cells per

millilitre), in RPMI supplemented with 10% FBS, Pen/Strep (100 U/ml) and 0.5 ng/ml M-CSF (Peprotech, 300-25) in order to derive monocyte-derived macrophages (MDMs). On day 3, half of the media was removed and replaced with fresh media to foster proliferation.

Primary murine bone marrow-derived macrophages (BMDMs) were prepared by flushing femurs of 3- to 6-month-old male and female wild-type C57BL/6 mice (Charles River UK). Red blood cells were lysed, and BMDMs were generated by culturing the resulting marrow cells in 70% DMEM (containing 10% (v/v) FBS, 100 U/ml penicillin, 100 μg/ml streptomycin) supplemented with 30% L929 mouse fibroblast-conditioned media for 7–10 days. BMDMs were seeded overnight at a density of $1 \times 10^6$ per millilitre in DMEM supplemented with 10% FBS, penicillin (100 U/ml) and streptomycin (100 μg/ml). Cells were maintained at 37°C, 5% $CO_2$ for all experiments.

Cells were primed for either 10 min or 4 h with LPS (1 μg/ml) as indicated. Following this, the priming stimulus was removed and cells were treated with one of the following inflammasome activators: Nigericin (10 μM, 45 min); Adenosine triphosphate (ATP, 5 mM, 1 h); Calcium pyrophosphate dehydrate (CPPD, 250 μg/ml, 3 h); dA:dT (as indicated) and flagellin (667 ng/ml, 4 h). Note that both Poly (dA:dT) and flagellin are incubated with the cells together with the transfection reagent Lipofectamine 3000, following the manufacturer's instructions. For non-canonical inflammasome activation, BMDMs were primed for 4 h with Pam3CSK4 (100 ng/ml). BMDMs were then incubated in serum-free DMEM containing either DMSO (0.1%), P22077 (2.5 or 10 μM) or MCC950 (1 μM) for 15 min before stimulation with LPS transfected with Lipofectamine 3000 (2 μg/ml, 24 h). P22077, HBX19818, P5091 and MCC950 were added, at the indicated doses, 15 min prior to the activation of the inflammasome, and were present during the treatment.

To assess the effect of high extracellular potassium, cells were primed for 4 h with LPS (1 μg/ml), before changing their media to either one containing physiological normal levels of solutes (147 mM NaCl, 10 mM HEPES, 13 mM glucose, 2 mM KCl, 2 mM $CaCl_2$ and 1 mM $MgCl_2$) or one containing a higher concentration of potassium (2 mM NaCl, 10 mM HEPES, 13 mM glucose, 145 mM of KCl, 2 mM $CaCl_2$ and 1 mM $MgCl_2$), both of which are buffered to pH 7.4. Following the media change, nigericin was added, as per normal.

## Quantitative real-time PCR

RNA was extracted from THP-1 and MDM macrophages using the PureLink RNA extraction kit from Thermo Fisher Scientific (#12183018A) and reverse-transcribed to cDNA using the High-Capacity RNA-to-cDNA™ Kit, both according to the manufacturer's instructions (#4387406). Specific primers for IL-1β (QT00021385), USP47 (QT00082201) and GAPDH (QT00079247) were purchased from Qiagen (QuantiTect Primer Assays). Primers for USP7, BRCC3, ASC, caspase-1 and IL-18 were designed using the primer express 3.0.1 software and obtained from Sigma. Primer sequences are as follows: hNLRP3-Fw: TGCCCGTCTGGGTGAGA; hNLRP3-Rv: CCGGTGCTCCTTGATGAGA; hCasp-1-Fw: ATACCAAGAACTG CCCAAGTTTG; hCasp-1-Rv: GGCAGGCCTGGATGATGA; hASC-Fw: GCCAGGCCTGCACTTTATAGA; hASC-Rv: GTTTGTGACCC TCGCGATAAG; hIL-18-Fw: AAGGAAATGAATCCTCCTGATAACA;

hIL-18Rv: CCTGGGACACTTCTCTGAAAGAA; hUSP7-Fw: CCAGTG CAATGCTGAATCTGA; USP7-Rv: ACGACGACTGAACGACTTTTC AT; BRCC3-Fw: GGAAATGCGCACAGTCCAA; BRCC3-Rv: CACGGTC CTTTCTCTTGTCAGA. qPCR was performed using Power SYBR® Green PCR Master Mix (Applied Biosystems, 4385618) and QuantStudio 12K Flex (Applied Biosystems). Gene expression was calculated using the ΔΔCt method. Data were normalised to expression levels of the housekeeping genes GAPDH and RPL37A for THP-1, and RPL37A and GNB2L1 for MDMs [60,61] across each treatment, and fold change was expressed relative to basal RNA levels from untreated cells.

## Cell death measurement

Cell death was measured using quantitative assessment of lactate dehydrogenase (LDH) levels in cell supernatants, after a centrifugation step of 5 min at 500 ×g at 4°C, to remove any dead/floating cells. CytoTox 96® Non-Radioactive Cytotoxicity Assay (Promega, G1780) was used, following the manufacturer's instructions. Plates were read at 490 nm and results shown as percentage of LDH release relative to the total cells lysed.

## Western blot

Cells were lysed for at least 20 min on ice using a RIPA lysis buffer (50 mM Tris–HCl, pH 8, 150 mM NaCl, 1% NP-40, 0.5% sodium deoxycholate and 0.1% sodium dodecyl sulphate, SDS), supplemented with a protease inhibitors cocktail, 20 mM NEM and 1 mM PMSF. Lysates were then centrifuged at 21,000 ×g for 10 min to remove the insoluble fraction. Protein concentration was measured by BCA assays (Thermo Scientific Pierce, 23225), following the manufacturer's instructions, and an equal amount of protein was loaded into each lane.

Cell supernatants were concentrated prior to Western blotting. Firstly, supernatants were centrifuged at 500 ×g for 5 min to remove dead cells before subsequently being concentrated with 10 kDa MW cut-off filters (Amicon, Merck Millipore), following the manufacturer's guidelines.

Supernatants and whole-cell lysates were diluted in Laemmli buffer containing 1% 2-mercaptoethanol, heated at 95°C for 10 min and resolved by SDS–PAGE. Separated proteins were transferred onto nitrocellulose membranes and blocked in 5% Milk TBS-Tween (0.1%) for 1 h at room temperature and incubated overnight with the specific primary antibody in blocking buffer. The following day, membranes were washed three times in TBS-Tween (TBST, 0.1%) for 10 min each and subsequently incubated for 1 h at room temperature with a horseradish peroxidase-conjugated secondary antibody. Membranes were subsequently washed again, as before, before being visualised using Clarity™ Western ECL Blotting Substrate (Bio-Rad, 1705061) in a ChemiDoc™ MP Imager (Bio-Rad).

## NLRP3 oligomerisation using native gels

For 1D Blue-Native PAGE (BN-PAGE), $1 \times 10^6$ primary WT BMDMs were seeded overnight into 12-well plates. LPS-primed (1 μg/ml, 4 h) BMDMs were incubated in serum-free DMEM containing either DMSO (0.1%) or P22077 (2.5 μM or 10 μM) for 15 min before

stimulation with ATP (5 mM, 30 min). Cells were lysed in a native lysis buffer (20 mM Bis-Tris, 500 mM 6-aminocaproic acid, 20 mM NaCl, 10% (v/v) glycerol, 0.25% (w/v) digitonin, 0.25% (v/v) Triton X-100, 0.5 mM $Na_3VO_4$, 0.5 mM NaF, 1 mM PMSF, protease inhibitor cocktail, pH 7.0) for 30 min at 4°C. Insoluble debris was pelleted by centrifugation at 20,000 ×$g$ for 30 min at 4°C. Soluble lysates were separated by isoelectric focusing using Coomassie brilliant blue G-250 on 3-12% NativePAGE Bis-Tris gels (Thermo Fisher) as described previously [62]. Given the absence of stained markers for native gels, the size of markers was estimated following separation range from manufacturer [1,236 kDa (16% down from top), 480 kDa (50% down from top) and 66 kDa (80% down from top)]. Proteins were then denatured by incubating the gel in 10% SDS for 10 min before semi-dry transfer and immunoblotting with indicated antibodies.

### NLRP3 ubiquitination analysis

In order to analyse the ubiquitination status of NLRP3 in human macrophages, we used THP-1 cells. Cells were plated in 10-cm dishes (10 ml at a concentration of $1 \times 10^6$ cells) by treatment with 50 nM PMA, overnight. The following morning, media were replaced by fresh complete RPMI and the cells were left for 24 h before subsequent treatments.

NLRP3 was immunoprecipitated using Protein G Dynabeads as per manufacturer's instructions. Briefly, cells were lysed in a NP-40 buffer, supplemented and processed as described above. Protein concentration was determined by BCA assay, and equal amounts of total protein (1.5 mg) were pre-cleared with the IgG-Dynabeads for 1 h at 4°C on a rotator. After that, lysates were collected and incubated with 1 µg of NLRP3 antibody or non-specific IgG2 per sample at 4°C on a rotator overnight. The next day, 20 µl of IgG-Dynabeads were added to the sample–antibody mix and incubated for 1 h at 4°C on a rotator. Beads were washed three times with NP-40 buffer, before being eluted in 1× Laemmli buffer at 95°C for 10 min.

Eluate from the NLRP3 IP was loaded, in equal volumes, into a pre-cast 3-8% Tris-acetate gel (NuPAGE™, EA0375, Thermo Fisher Scientific). Proteins were transferred onto methanol-activated 0.45 µm PVDF membranes (GE Healthcare Amersham™ Hybond™, Thermo Fisher Scientific, 15269894). Membranes were subsequently incubated for 30 min at 4°C in denaturing buffer (6 M guanidine/HCl, 20 mM Tris–HCl, pH 7.5 and 5 mM 2-mercaptoethanol). Following this, membranes were washed three times in TBST and blocked in 1% BSA in TBST. Membranes were then incubated overnight with anti-Ub-HRP or anti-Ub-K63-HRP in 1% BSA in TBST and visualised as per normal.

### DUB activity-based probe labelling

Cells were plated and treated as per the experiment and washed three times with ice-cold PBS and lysed in labelling buffer (50 mM Tris pH 7.4, 250 mM sucrose, 5 mM $MgCl_2$, 0.5% Triton X-100, 0.5 mM EDTA). Lysates were vortexed and centrifuged at 14,000 ×$g$ for 10 min, 4°C. Protein concentration was determined by BCA and normalised to 2 mg/ml with the labelling buffer. The TMR-Ub-VME probe (UbiQ) was added at a final concentration of 0.5 µM and incubated with the lysate for 20 min at 37°C.

Following this, the reaction was quenched with SDS sample buffer, and SDS–PAGE was performed to detect shifts in molecular weight. For both USP7 and USP47, homemade 6% Tris-glycine gels were used for this purpose.

### Enzyme-linked immunosorbent assay (ELISA)

Levels of both human and murine IL-1β and human IL-18 were measured in the cell supernatants using ELISA kits from R&D Systems (human IL-1β, DY201; mouse IL-1β, DY401) and e-bioscience (IL-18, BMS267/2MST). TNF-α and IL-6 were measured in cell supernatants using ELISA kits from Thermo Fisher Scientific (TNF-α, 88-7346-88; IL-6, 88-7066-88). ELISAs were performed following the manufacturer's instructions.

### ASC speck oligomerisation assay

Cell lysates and supernatants were assayed for ASC speck formation as previously published [63]. For the supernatants, the media were centrifuged at 500 ×$g$ for 5 min to remove dead cells and subsequently concentrated with 10 kDa MW cut-off filters (Amicon, Merck Millipore), following the manufacturer's guidelines. The crosslinker DSS (disuccinimidyl suberate, Thermo Fisher Scientific, 21658) was added to a final concentration of 2 mM and incubated at 37°C for 20 min.

For the lysates, cells were lysed in Buffer A (20 mM HEPES-KOH, pH 7.5, 10 mM KCl, 1.5 mM $MgCl_2$, 1 mM EDTA, 1 mM EGTA, 320 mM sucrose and protease inhibitor cocktail). Lysates were then passed through a 21-gauge needle by syringing 30 times. Lysates were then centrifuged at 300 ×$g$ for 8 min. Supernatants were then transferred carefully to an empty Eppendorf tube without disturbing the nuclear pellet and diluted with 1 volume of CHAPS buffer (20 mM HEPES-KOH, pH 7.5, 5 mM $MgCl_2$, 0.5 mM EGTA, 0.1% CHAPS and protease inhibitor cocktail) and centrifuged at 2,400 ×$g$ for 8 min in a refrigerated microcentrifuge. Supernatants were discarded, and the pellets were washed twice with 0.5 ml ice-cold PBS. Washed pellets were then re-suspended in 30 µl CHAPS buffer, and DSS crosslinker was added to a final concentration of 2 mM and incubated at 37°C for 20 min. Reactions were quenched by adding Laemmli buffer and boiling prior to performing SDS–PAGE.

### Immunofluorescence and ASC speck quantification

THP-1 cells were plated and treated as described above on glass coverslips. Cells were fixed with 4% paraformaldehyde in PBS for 30 min. Subsequently, cells were permeabilised with 0.1% Triton X-100. A blocking step of 1 h using 5% BSA in PBS (block solution) was used before incubation with rabbit anti-ASC (1:50) in block solution for 1 h. Coverslips were then washed 5 times in PBS. ASC antibodies were detected by incubation with Alexa Fluor 488-conjugated donkey anti-rabbit antibody (1:200) in blocking solution for 1 h. The coverslips were washed again 5 times in PBS and finally in distilled water before being dried and mounted onto a glass slide using ProLong Gold mounting medium containing DAPI (Invitrogen). Images were taken with an Olympus BX51 upright microscope using a 10/0.50 Plan Fln objective and captured using a Coolsnap EZ camera (Photometrics) through MetaVue Software (Molecular Devices). To quantify the extent of

speck formation, the percentage of cells that contained an ASC speck was calculated. Cells from 10 different fields were counted, and the average was counted as one of each of the independent experiments. Images were analysed using ImageJ (rsb.info.nih.gov). The data are expressed as the percentage of ASC specks per number of cells per field.

### Lentiviral constructs

The inducible lentiviral construct FgH1tUTG was a gift from Marco Herold [44] (Addgene plasmid #70183), and lentiCas9-Blast was a gift from Feng Zhang (Addgene plasmid #52962) [64]. Packaging plasmids psPAX2 and pMD2.G were a gift from Didier Trono (Addgene plasmid #12260 and #12259).

### sgRNA design and expression analysis

The WTSI Genome Editing CRISPR design software [65] was used for the design of sgRNA (http://www.sanger.ac.uk/htgt/wge/). To clone individual sgRNAs, 24-bp oligonucleotides containing the sgRNA sequences were synthesised (Sigma). They included a 4-bp overhang for the forward (TCCC) and complementary reverse (AAAC) oligos to enable cloning into the Bsmb-I site of the lentiviral construct. sgRNA sequences are as follows: USP7-g1: 5′-GGACACAA CACCGCGGAGG; USP7-g2: 5′-GATGGACACAACACCGCGG; USP47-g2: 5′-ATAGGTCCGCTTCCAAGAGA; USP47-g3: 5′-GTCGAGAAGT GACTTGTCAC.

### Virus production and transduction of cell lines

HEK293T cells ($3.5 \times 10^5$ per well in a six-well plate) were plated for 24 h. Lipofectamine 2000 was used following the manufacturer's instructions. In short, 8 µl Lipofectamine, 1.2 µg pMD2.G, 0.4 µg psPAX2 and 1.5 µg of our vector of interest (either lentiCas9 or vector containing sgRNA) were used per reaction to transfect HEK293T cells. The following day, the media were replaced and cells were further incubated for 2 days. Supernatants were then filtered with a 0.45-µm filter to obtain a cell-free extract of viral particles. Viral particles containing our vector of interest were used to transduce $5.0 \times 10^4$ THP-1 cells, with 8 µg/ml polybrene. Cells, together with both the viral particles and polybrene, were centrifuged at $1,000 \times g$ for 1 h at 30°C. Pelleted cells were then re-suspended in fresh complete RPMI and plated in a 12-well plate.

### Generation of stable cell lines

To obtain Cas9-stably expressing THP-1 cells, THP-1 cells transduced with lentiCas9-Blast were selected with blasticidin (10 µg/ml) for 10 days. Single-cell clones were selected by dilution into 96-well plates, and Cas9 expression was assessed by Western blot (using the anti-CRISPR/Cas9 antibody, EPR19619-92, ab210752; data not shown).

To obtain guide-expressing THP-1 cell lines, THP-1 Cas9-expressing cells were transduced with the FgH1tUTG vector containing the guide of interest. Virally transduced cell lines were sorted for eGFP-positive cells using a BD Influx cell sorter (top 5% eGFP-expressing population).

### Doxycycline treatment

To induce expression of gRNA, cell lines were treated with doxycycline hyclate (Sigma-Aldrich, D9891) at 1 µg/ml for 3 days. After treatment, cells were maintained as per usual in the absence of doxycycline.

### Statistical analysis

Statistical analysis was performed using GraphPad Prism 7 software. Differences between two groups were identified using *t*-test analysis (qPCR data, ASC speck formation and IL-1β release in NLRP3 KO mice). Differences between three or more groups were identified using either one-way ANOVA with the Greenhouse–Geisser correction and with the *post hoc* Dunnett's multiple comparisons test, or two-way ANOVA with *post hoc* Tukey's test for multiple comparisons.

**Expanded View** for this article is available online.

### Acknowledgements

This work was supported by a Sir Henry Dale Fellowship jointly funded by the Wellcome Trust and the Royal Society (Grant number 104192/Z/14/Z) to G.L-C, by the Manchester Collaborative Centre for Inflammation Research, a joint initiative of the University of Manchester, AstraZeneca, and GlaxoSmithKline and by a Medical Research Council (MRC) grant (MR/N029992/1) to D.B. The Bioimaging Facility microscopes used in this study were purchased with grants from BBSRC, Wellcome Trust and the University of Manchester Strategic Fund. The authors thank Dr Gareth Howell (Manager, Manchester University Core Flow Cytometry Facility) for technical assistance with cell sorting.

### Author contributions

PP-R, JDW, FM-S, AV, JG, CP and GL-C performed experiments and interpreted results. PP-R and JDW contributed equally to this work. JDW performed IL-1β/IL-18 release experiments in MDMs, ELISAs, Western blotting, LDH assays, ABP assays, ASC oligomerisation and cell fractionation assays and CRISPR/Cas9 experiments. PP-R performed and analysed IL-1β/IL-18 release experiments in MDMs, ELISAs, Western blotting and LDH assays. AV performed qRT–PCRs. JG performed all work with murine BMDMs. CP performed IL-1β release assays in THP-1 cells. GL-C performed ASC speck immunofluorescence assays, qRT–PCRs and CRISPR/Cas9 experiments. DB contributed to experimental design of murine work and provided intellectual input. GL-C conceived and oversaw the project and wrote the first draft of the manuscript. PP-R, JDW and GL-C edited and approved the final version of the manuscript.

### Conflict of interest

The authors declare that they have no conflict of interest.

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
