## [Review Process File · EMBO Reports]

USP7 and USP47 deubiquitinases regulate NLRP3 inflammasome activation

Pablo Palazon-Riquelme, Jonathan D Worboys, Jack. Green, Ana Valera, Fátima Martín-Sanchez, Carolina Pellegrini, David Brough, Gloria López-Castejón

Review timeline:

Submission date:	5 July 2017
Editorial Decision:	16 August 2017
Revision received:	15 January 2018
Editorial Decision:	14 February 2018
Revision received:	24 May 2018
Editorial Decision:	25 June 2018
Revision received:	3 July 2018
Editorial Decision:	25 July 2018
Revision received:	2 August 2018
Accepted:	7 August 2018

Transaction Report:

1st Editorial Decision

16 August 2017

Thank you for the submission of your research manuscript to our journal. We have now received the full set of referee reports that is copied below.

As you will see, while all referees in principle agree on the potential interest of your findings, they also raise a number of - often overlapping - issues that would need to be addressed before publication. In particular, more convincing evidence will be required to support the claim that USP7/47 target ASC. All referees request further biochemical evidence that endogenous ASC is (de-)ubiquitinated.

Given these constructive comments, we would like to invite you to revise your manuscript with the understanding that the referee concerns (as detailed above and in their reports) must be fully addressed and their suggestions taken on board. Please address all referee concerns in a complete point-by-point response. Acceptance of the manuscript will depend on a positive outcome of a second round of review. It is EMBO reports policy to allow a single round of revision only and acceptance or rejection of the manuscript will therefore depend on the completeness of your responses included in the next, final version of the manuscript.

Revised manuscripts should be submitted within three months of a request for revision; they will otherwise be treated as new submissions. Please contact us if a 3-months time frame is not sufficient for the revisions so that we can discuss the revisions further.

Supplementary/additional data: The Expanded View format, which will be displayed in the main HTML of the paper in a collapsible format, has replaced the Supplementary information. You can submit up to 5 images as Expanded View. Please follow the nomenclature Figure EV1, Figure EV2

etc. The figure legend for these should be included in the main manuscript document file in a section called Expanded View Figure Legends after the main Figure Legends section. Additional Supplementary material should be supplied as a single pdf labeled Appendix. The Appendix includes a table of content on the first page, all figures and their legends. Please follow the nomenclature Appendix Figure Sx throughout the text and also label the figures according to this nomenclature. For more details please refer to our guide to authors.

Regarding data quantification, can you please specify the number "n" for how many independent experiments were performed, the bars and error bars (e.g. SEM, SD) and the test used to calculate p-values in each respective figure legend? As also pointed out by the referees, only the data obtained from independent biological replicates can be used to calculate statistics.

Moreover, we now strongly encourage the publication of original source data with the aim of making primary data more accessible and transparent to the reader. The source data will be published in a separate source data file online along with the accepted manuscript and will be linked to the relevant figure. If you would like to use this opportunity, please submit the source data (for example scans of entire gels or blots, data points of graphs in an excel sheet, additional images, etc.) of your key experiments together with the revised manuscript. Please include size markers for scans of entire gels, label the scans with figure and panel number, and send one PDF file per figure or per figure panel.

- a complete author checklist, which you can download from our author guidelines (<http://embor.embopress.org/authorguide#revision>). Please insert page numbers in the checklist to indicate where the requested information can be found.
- a letter detailing your responses to the referee comments in Word format (.doc)
- a Microsoft Word file (.doc) of the revised manuscript text
- editable TIFF or EPS-formatted figure files in high resolution
- a separate PDF file of any Supplementary information (in its final format)
- all corresponding authors are required to provide an ORCID ID for their name. Please find instructions on how to link your ORCID ID to your account in our manuscript tracking system in our Author guidelines (<http://embor.embopress.org/authorguide>).

As part of the EMBO publication's Transparent Editorial Process, EMBO reports publishes online a Review Process File to accompany accepted manuscripts. This File will be published in conjunction with your paper and will include the referee reports, your point-by-point response and all pertinent correspondence relating to the manuscript.

I look forward to seeing a revised version of your manuscript when it is ready. Please let me know if you have questions or comments regarding the revision.

REFEREE REPORTS

Referee #1:

Recent studies have suggested that inflammasome assembly is regulated by the ubiquitin system and that both ubiquitination and deubiquitination events are involved in this process. Here the authors demonstrate that inhibition of the homologous DUBs USP7 and USP47 blocks NLRP3

inflammasome activation in human macrophages using specific inhibitors as P22077, P0590 or HBX19818. As some DUBs regulates the NF- κ B pathway, the authors show that the transcriptional inflammasome priming is not required for USP7/47 mediated inflammasome activation. Interestingly, they observed that P22077 impairs ASC oligomerization and speck formation, and using BRET, they show that this inhibitor reverses ASC deubiquitination after NLRP3-inflammasome activation. Moreover, in response to danger signals, USP7/47 are post-transcriptionally regulated. Finally, they report that while NLRP3-inflammasome activation was normal in USP7 deficient THP-1, it was reduced in USP47 deficient cells, but this inhibitory effect was potentiated when double USP7/USP47 deficient cells were used, suggesting that both DUBs contribute to NLRP3-inflammasome activation.

In this manuscript, the authors mainly focused on NLRP3 inflammasome activation. In figure 3A, after transfection of Poly (dA:dT), the use of P22077 also suggests that USP7/47 inhibition prevents AIM2-inflammasome activation. As, both NLRP3- and AIM2-inflammasome require the adaptor ASC, the authors have shown, using BRET, that P22077 inhibits ASC deubiquitination and thus inflammasome activation. As ubiquitination/deubiquitination can be assessed in biochemistry using immunoprecipitation followed by ubiquitin detection in WB, I guess that the authors should show, to confirm the BRET, that during NLRP3-inflammasome activation, ASC is deubiquitinated, a process that is prevented after USP7/47 inhibition. Likewise, as the authors have made USP7/47 deficient THP-1 using gene editing, these cells should be used to assess in biochemistry the ubiquitination state of ASC during NLRP3 inflammasome activation. Finally, previous studies have demonstrated in biochemistry that NLRP3 has to be deubiquitinated for its activation, so whether USP7/47 may affect its ubiquitination state should also be explored to confirm that both DUBs are specific for ASC.

Other comments.

In figure 3B, I do not understand why the transfection of poly (dA:dT) does not promote cell death as AIM2-inflammasome triggers caspase-1 activation leading to gasdermin D cleavage and ensuing pyroptosis.

In Figure 6, poly (dA:dT) should be transfected in control or USP7/47 deficient THP1 after LPS priming to assess IL-1b release and cell death in order to confirm the involvement of both DUBs in AIM2 inflammasome.

Referee #2:

In their manuscript, Palazon-Riquelme et al use small molecule inhibitors of USPs and show their effect on NLRP3 and AIM2 inflammasomes. This study builds on recent work on DUSPs (BRCC3, USP50 etc) in inflammasome activation. They suggest that these effects are via USP7 and USP47. The ubiquitin protease activities of these, or other, USPs is required for ASC oligomerization and inflammasome activation. They use a BRET2-based assay to show changes to ASC ubiquitylation which correlate with its activation status. Activity based probes are used to show increased USP activity in cells undergoing inflammasome activation. The authors generate inducible-CRISPR based knockouts in THP1 cells and show that loss of USP7 and USP47 leads to reduced inflammasome activation with nigericin. Overall, these findings are interesting, of interest to the field, and describe new assays. However, a number of controls are missing and some of the inferences are based on assays that need to be developed further. One major concern that they claim that USPs act on ASC but do not show changes to ubiquitylation of endogenous ASC in MDMs or THP1 which they have used extensively for assays in their study.

The following major concerns need to be addressed.

1. In Fig 1 the authors claim the drugs block NLRP3 inflammasome activation, but do not show a single caspase-1 activation blot. Same is true for Fig 2. The authors should show caspase-1 p10 or p20 bands to confirm the impact of drugs in MDMs on caspase-1 inflammasome activation. Also, full blots for IL-1b and IL-18 should be shown rather than crops of the 15-20 kDa regions of blots.
2. The authors claim, by testing the NLRP3 and AIM2 inflammsomes, that the common defect lies in ASC oligomerization. They should test additional inflammasomes as reagents for activating Pyrin (C3 toxin), NLRC4 (flagellin) and caspase-11/4 (cytosolic LPS) are readily available. More importantly, the authors should test the effect of ATP as a natural endogenous activator of NLRP3 to which MDMs are particularly responsive.

3. Authors claim that USP inhibitors do not affect transcription but have only shown RT-PCR data for several genes (Fig 2A & EV) with LPS priming in the absence of inhibitors - the same should be shown with the inhibitors.
4. Blots in Fig 2B also need a (-)LPS control to assess induction of NLRP3 and pro-IL1b without and with inhibitors.
5. What is the difference between experiments in Fig 3 panels C and D? They look almost the same experiments. The dimer in C is not really visible, but is marked. In the same Fig, why aren't cells dying from DNA transfection as expected for AIM2 inflammasomes? These experiments also need caspase-1 blots. Based on experience with LDH assays, which are less sensitive, I feel this could be an issue of sensitivity. Could the authors try propidium iodide uptake as a surrogate for gasdermin-D mediated pores that form before osmotic lysis? PI (or related that can be used with nucleic acid activators) fluorescence based assay is more sensitive and might unmask small differences between the inhibitor treatment and non-treatment. I am less puzzled by the lack of impact on LDH release in response to CPPD, which might kill cells through other necrotic processes.
6. If indeed it turns out there is no difference in cell lysis (or not), the authors should test pooled supernatant+pellet blot to test whether processed cytokines are not being released. As shown currently (ELISA and westerns on supernatants), the authors haven't ruled out the possibility that there may be processed intracellular IL-1b/IL-18.
7. The BRET assay is quite interesting and should be developed further. However, I have several questions about the assay which could be addressed by additional controls. For example, I would like to see that their assays show defects when using ASC mutants that do not oligomerise or bind to NLRP3. My major concern is that the authors use HEK293 cells for these experiments, and these cells are known to not express NLRP3. How is Nigericin leading to activation in this system? Reconstitution of HEK cells with various inflammasome proteins for mutational studies has been done before and this is still a valid biochemical assay system. However, the authors should note that nigericin causes golgi collapse and other toxicity issues even in the absence of NLRP3. In order to show specificity of their BRET2 assay, therefore, they must carry out assays with co-transfected NLRP3 (and ideally inactive mutants lacking the PYD region) to establish that their assay is a faithful reporter of signal-driven ASC assembly.
8. In continuation with point 6, the authors should show changes to ubiquitylation of endogenous ASC in macrophages due to USP inhibitor treatments so as to correlate findings from the BRET assay. The BRET2 assay in HEK293 is not a sufficient measure of what might be going on in MDMs.
9. The ABP approach, which is again quite interesting, would seem to suggest that the compounds also inhibit other USPs, although the authors discuss that the irreversibility of their probes. In such a scenario, it would be good to try nigericin treatment in excess K⁺ to block activation and show that the probes can reveal specific activation with nigericin ie. the K⁺ efflux required for NLRP3 activation somehow activates USPs? Have the authors tested that the bands correspond to USP7/47 by trying similar assays in the gRNA-based KO cells?
10. The authors should at least additionally test AIM2 inflammasomes in the CRISPR KO cells and show that endogenous ASC ubiquitylation is altered in these cells.
11. Statistics: the authors must clarify that when they mean n=10 for ELISAs etc (for example in Fig legends), these are 10 means from biologically independent experiments and do not include technical replicates. Only means from independent experiments should be used for statistical calculations.

Minor points.

1. pg 3 - middle para - authors mean large number of DUBs (rather than high number)
2. pg 4 line 10 from top (line numbers would have been useful throughout) - in this instance and other places where the authors discuss sequence relatedness or similarities, they use the term 'homology'. They must refrain from using this incorrect term even if widely used. Homology means related by descent - a sequence descended from an ancestral gene or not (gene is a daughter of a parent or not) - there is no such thing as partial homology. However, there can be partial sequence similarities which can be expressed quantitatively simply by stating % sequence similarity. They should state % protein sequence similarity between the USPs throughout and not use the word homology.
3. pg 9 - line 4 - different instead of deferent.

Referee #3:

This paper investigates two deubiquitinase enzymes in the regulation of NLRP3 inflammasome responses, and shows that USP7 and USP47 are both activated in response to NLRP3 triggering. They are implicated in the response by use inhibitors but importantly also CRISPR knockout (Figure 6) and an activity assay confirms activation of the enzymes under inflammasome triggering conditions (Figure 5). Figure 4 is an attempt to show ASC as a target of the deubiquitinases, but it fails to convince. The authors have attempted a BRET technique with overexpression of components in HEK cells. It is not clear why they have done this, without including any data on endogenous ASC ubiquitination in macrophages. Figure 4 is the major problem with the paper, as panel B gives no confidence in any of the data of this figure. (Please note that although I would rate the technical quality of most of the paper as high, I would rate this figure as unacceptable). Inspection of reference 57 on which they base this technique gives clear saturation curves that are at a distinctly higher signal than the negative control that increases linearly. The data presented here show random noise not significantly above the negative control, and curve fitting to this data is not realistic. The authors themselves do not seem confident enough of this data to include it in the abstract. Otherwise, most of the paper is well executed, but at this stage should conclude that these deubiquitinases promote NLRP3 inflammasome activity, through an unknown target.

Specific Comments

Figure 4 should be omitted. If you have any data from macrophages to show effects on endogenous ubiquitination, that would be much superior, and would raise the level of the paper substantially. In the absence of that, the target of the deubiquitinases remains not established.

Figure 2. Panel B is not what the legend says. It shows nigericin treatment apparently, and not LPS treatment. If it was +/- LPS treatment, we would see pro-IL-1b induced. I am not sure what this is showing.

Figure 3. There is no guarantee within this figure that the chemically transfected polydA-dT (that is giving a very low response) is actually recognized by AIM2, and this is only presumed by extension from mouse work. However, this response has not been well explored in human cells. This would not be so important except that effects on both NLRP3 and AIM2 pathways are used as a justification that the DUBs are acting on something affecting ASC, and not something further upstream in the NLRP3 pathway.

Some legends say n=11 individual experiments. Others just say "n=11" and the level at which replication is done is not clear. Please clarify in every legend. For example, n=8 donors or n=8 experiments etc.

If the authors are convinced that ASC ubiquitination is important, then this question is relevant. If DUBs were to regulate the polymerization of ASC, then this infers that a substantial proportion of ASC should be ubiquitinated, and this should be seen just on a western blot for ASC. Are there higher bands on total ASC blots that are consistent with ubiquitination, and that are altered following inflammasome triggering? Or otherwise, if ASC ubiquitination is only of a small minority of the protein, how would this have an important effect on the behavior of the bulk ASC?

Figure 3C and D. What is the "ASC input"? The Figure legend does not fully describe what has been done here, as the methods state that you are actually looking at the ASC speck pellets isolated by centrifugation. In Figure 3C, this should not say lysates, but should say ASC speck pellets isolated by 2400g centrifugation. So at what stage is the "ASC input" taken? It is obviously not the ASC speck fraction without crosslinking, as ASC is in all samples in panel C. Is it the lysate after 300g spin? With the lysis conditions used (isotonic, and 21G needle), I expect pyroptotic cells would readily lyse, but the control cells would not. Did you check that cells actually lysed with these conditions? Also in Fig 3D, why are there two panels for ASC crosslinked- are they different exposures?

Minor points

Figure 3F. How you got error bars and stats here is not clear, and should be explained in the legend, not just in the methods. Provide n and an explanation please.

Discussion - "requires nucleate on-induced polymerization". Is this supposed to be "nucleation"?
 Methods- cells and treatments - problem with exponent on cell concentration; also "plaquets"- do you mean platelets?

Although I think this data should be taken out, note that filters for luciferase and GFP are listed at the same wavelength. Also "the milliBRET units are calculated as the percentage of BRET signal relative to basal signal, and results are expressed as the average of mBU after nigericine treatment". What is meant by the "basal signal"? How can anything called a "milli-unit" be a percentage? The processing of data for this is not at all easy to understand.

Figure 1H legend "PS-primed" typo

1st Revision - authors' response

15 January 2018

Referee #1:

Recent studies have suggested that inflammasome assembly is regulated by the ubiquitin system and that both ubiquitination and deubiquitination events are involved in this process. Here the authors demonstrate that inhibition of the homologous DUBs USP7 and USP47 blocks NLRP3 inflammasome activation in human macrophages using specific inhibitors as P22077, P0590 or HBX19818. As some DUBs regulates the NF- κ B pathway, the authors show that the transcriptional inflammasome priming is not required for USP7/47 mediated inflammasome activation. Interestingly, they observed that P22077 impairs ASC oligomerization and speck formation, and using BRET, they show that this inhibitor reverses ASC deubiquitination after NLRP3-inflammasome activation. Moreover, in response to danger signals, USP7/47 are post-transcriptionally regulated. Finally, they report that while NLRP3-inflammasome activation was normal in USP7 deficient THP-1, it was reduced in USP47 deficient cells, but this inhibitory effect was potentiated when double USP7/USP47 deficient cells were used, suggesting that both DUBs contribute to NLRP3-inflammasome activation.

In this manuscript, the authors mainly focused on NLRP3 inflammasome activation. In figure 3A, after transfection of Poly (dA:dT), the use of P22077 also suggests that USP7/47 inhibition prevents AIM2-inflammasome activation. As, both NLRP3- and AIM2-inflammasome require the adaptor ASC, the authors have shown, using BRET, that P22077 inhibits ASC deubiquitination and thus inflammasome activation. As ubiquitination/deubiquitination can be assessed in biochemistry using immunoprecipitation followed by ubiquitin detection in WB, I guess that the authors should show, to confirm the BRET, that during NLRP3-inflammasome activation, ASC is deubiquitinated, a process that is prevented after USP7/47 inhibition.

We thank the reviewer for their comments here and we agree that biochemical evidence to support the claim that ASC is deubiquitinated was an important next step. We did attempt to purify endogenous ASC from THP-1 cells, and tested four different antibodies in a RIPA lysate for this purpose:

- ASC B-3 sc-514414,
- ASC F-9 sc-271054,
- ASC N-15 sc-22514-R from Santa Cruz
- ASC (AL177) AG-25B-0006-C100 from Adipogen

Unfortunately, none of the antibodies we tested provided an efficient immunopurification, which made an assessment of the ubiquitination state very difficult. To that end, we used THP-1 cells expressing ASC-mRFP and utilised the RFP-TRAP system for purification. This allowed for a far superior IP that we were then able to blot for Ub. By using this system we have been able to show that ASC, in a macrophage cell line, is deubiquitinated upon nigericin treatment that is reversed in the presence of P22077. This data is now part of a revised figure 4 (Fig. 4E).

Likewise, as the authors have made USP7/47 deficient THP-1 using gene editing, these cells should be used to assess in biochemistry the ubiquitination state of ASC during NLRP3 inflammasome activation.

We agree that the ubiquitination state of ASC in the CRISPR/Cas9-edited cells would be an important experiment. As we have been unable to IP endogenous ASC, we were unfortunately unable to do this experiment.

Finally, previous studies have demonstrated in biochemistry that NLRP3 has to be deubiquitinated for its activation, so whether USP7/47 may affect its ubiquitination state should also be explored to confirm that both DUBs are specific for ASC.

We have now included more evidence to support the claim that USP7 and USP47 is not specifically blocking the NLRP3 inflammasome. We now show that the USP7/47 inhibitor is able to block NLRP3, AIM2 and NLRC4 activation, and have confirmed the results for both the NLRP3 and AIM2 inflammasome in USP7/47 deficient cells with CRISPR/Cas9. Additionally, we see that P22077 inhibition impairs ASC oligomerisation and speck formation. Given that ASC oligomerisation can occur independently of the presence of NLRP3 (Compan et al., 2015), and that we detect very similar levels of inhibition with P22077 in the NLRP3 KO mice compared to WT BMDMs (now Fig. EV4) we think it is highly unlikely that NLRP3 is the target for deubiquitination. Thus we focused our efforts on studying ASC ubiquitination, yet we agree we cannot claim that these DUBs are solely specific to ASC and have rewritten the text to emphasis this.

Other comments.

In figure 3B, I do not understand why the transfection of poly (dA:dT) does not promote cell death as AIM2-inflammasome triggers caspase-1 activation leading to gasdermin D cleavage and ensuing pyroptosis.

IL-1 β release can occur when levels of LDH are very low or non-detectable, as it has been shown when cells are treated with the cryoprotective agent glycine (Fink and Cookson, 2006). It has also been shown that IL-1 β can be released prior to the onset of cell death (Evavold et al., 2017). Detection of levels of cell death by LDH will depend on the time point at which IL-1 β is measured. Initially we performed our experiments using 16 hours of stimulation, which led to low levels of IL-1 β release (mean of 120 pg/mL) and non-detectable changes in LDH. We have now repeated these experiments with a longer exposure to dAdT (now 24 hours) and we detect higher levels of IL-1 β release (10-fold increase) as well as a detectable increase in LDH release after stimulation (updated in Fig 3A,B). The cell death achieved here was not blocked by P22077 and similar data was obtained when AIM2 activation was performed in mouse macrophages (Fig EV4). This indicates that USP7/47 inhibition affects the IL-1 β processing but not AIM2-derived cell death.

In Figure 6, poly (dA:dT) should be transfected in control or USP7/47 deficient THP1 after LPS priming to assess IL-1b release and cell death in order to confirm the involvement of both DUBs in AIM2 inflammasome.

We agree with the reviewer that this experiment was important to the paper and thank them for their suggestion. We have thus now performed these experiments (now Fig. EV7). In agreement with the inhibitor data discussed in the point above we observed reductions of IL-1 β release by Western blotting and by ELISA in the double USP7/USP47 deficient cells, but did not observe any effect on AIM2-induced cell death.

Referee #2:

In their manuscript, Palazon-Riquelme et al use small molecule inhibitors of USPs

and show their effect on NLRP3 and AIM2 inflammasomes. This study builds on recent work on DUSPs (BRCC3, USP50 etc) in inflammasome activation. They suggest that these effects are via USP7 and USP47. The ubiquitin protease activities of these, or other, USPs is required for ASC oligomerization and inflammasome activation. They use a BRET2-based assay to show changes to ASC ubiquitylation which correlate with its activation status. Activity based probes are used to show increased USP activity in cells undergoing inflammasome activation. The authors generate inducible-CRISPR based knockouts in THP1 cells and show that loss of USP7 and USP47 leads to reduced inflammasome activation with nigericin. Overall, these findings are interesting, of interest to the field, and describe new assays. However, a number of controls are missing and some of the inferences are based on assays that need to be developed further. One major concern that they claim that USPs act on ASC but do not show changes to ubiquitylation of endogenous ASC in MDMs or THP1 which they have used extensively for assays in their study.

We thank this reviewer for their encouraging comments. As we have mentioned in the reponse to the reviewer #1 (please see above), we agree that an assessment of ASC in the endogenous setting was important to the paper and to that end, we used THP-1 cells expressing ASC-mRFP and utilised the RFP-TRAP system for purification. This allowed for a far superior IP than to antibodies we had tested, enabling us to blot for Ub. By using this system we have been able to show that ASC, in a macrophage cell line, is deubiquitinated upon nigericin treatment that is reversed in the presence of P22077. This data is now part of a revised figure 4 (Fig. 4E).

The following major concerns need to be addressed.

1. In Fig 1 the authors claim the drugs block NLRP3 inflammasome activation, but do not show a single caspase-1 activation blot. Same is true for Fig 2. The authors should show caspase-1 p10 or p20 bands to confirm the impact of drugs in MDMs on caspase-1 inflammasome activation. Also, full blots for IL-1 β and IL-18 should be shown rather than crops of the 15-20 kDa regions of blots.

We agree that a more complete figure would include active caspase-1 blots as well as full blots for IL-1 β and IL-18 that have now been included in all relevant figures throughout the manuscript. As can be seen P22077 and USP7/USP47 deficiency blocks the cleavage and release of caspase-1.

2. The authors claim, by testing the NLRP3 and AIM2 inflammasomes, that the common defect lies in ASC oligomerization. They should test additional inflammasomes as reagents for activating Pyrin (C3 toxin), NLRC4 (flagellin) and caspase-11/4 (cytosolic LPS) are readily available. More importantly, the authors should test the effect of ATP as a natural endogenous activator of NLRP3 to which MDMs are particularly responsive.

We have now included data using the activators ATP and flagellin in mouse BMDMs, which demonstrate that P22077 also blocks the activation of these inflammasomes. Although human MDMs respond to ATP we observed a high variation in this response in MDMs - probably due to different levels of expression of P2X7R as well as the different haplotypes present in the human population (Fuller et al., 2009), which we have explained in the text (lines 121-124). We also tried the same protocol for NLRC4 activation, which we successfully employed in murine BMDMs, in human MDMs. Unfortunately we failed to activate this inflammasome, with no detectable levels of IL-1 β or IL-18 even after 24 hours of transfection.

3. Authors claim that USP inhibitors do not affect transcription but have only shown RT-PCR data for several genes (Fig 2A & EV) with LPS priming in the absence of inhibitors - the same should be shown with the inhibitors.

We agree with the reviewer that this was not sufficient, and have repeated this according to reviewers comments (lines 140-146; Fig. 2A). We observed no effect of

the drugs on mRNA levels of IL-18, ASC, caspase-1 or NLRP3. We did however observe a significant decrease in IL-1 β mRNA levels. Even though we observe a decreased transcriptional response of IL-1 β production, we do not see great differences in the total levels of pro-IL-1 β in P22077 treated cells, or in the CRISPR/Cas9-edited cells.

4. Blots in Fig 2B also need a (-)LPS control to assess induction of NLRP3 and pro-IL1b without and with inhibitors.

We thank the reviewer for this suggestion and have now included this in the figure (Fig. 2B).

5. What is the difference between experiments in Fig 3 panels C and D? They look almost the same experiments. The dimer in C is not really visible, but is marked.

We thank the reviewer for pointing this and have made steps to clarify this (lines 193-200). The data in question (which now form Fig. 4A and B) shows ASC oligomerisation in cell lysates (Fig. 4A) and in cell supernatants (Fig. 4B). Oligomerisation of ASC-within the cell is important for inflammasome activation and the processing and release of IL-1 β and IL-18. However it has also been shown that oligomeric-extracellular ASC is important in propagating inflammasome activation to neighbouring cells (Franklin et al., 2014; Baroja-Mazo et al., 2014) and hence why we believed it was important to also measure this in our experimental conditions. Additionally we have added the words 'Cell lysate' and 'Supernatant' above the corresponding blots, to make the origin of the inputs clearer. On reflection, we agree that the dimer in C is not visible, so we have now removed this label.

In the same Fig, why aren't cells dying from DNA transfection as expected for AIM2 inflammasomes? These experiments also need caspase-1 blots. Based on experience with LDH assays, which are less sensitive, I feel this could be an issue of sensitivity. Could the authors try propidium iodide uptake as a surrogate for gasdermin-D mediated pores that form before osmotic lysis? PI (or related that can be used with nucleic acid activators) fluorescence based assay is more sensitive and might unmask small differences between the inhibitor treatment and non-treatment. I am less puzzled by the lack of impact on LDH release in response to CPPD, which might kill cells through other necrotic processes.

As we have previously mentioned for the first reviewer's comment we have now addressed the issue of not detecting cell death in AIM2 inflammasome-activated cells. We agree that our issue was likely that of sensitivity and that by using a longer activation time with dAdT, we were able to detect LDH release, as well as higher levels of IL-1 β .

We agree that a more complete figure would include active caspase-1 blots as well as full blots for IL-1 β that have now been included (now as a standalone figure 3).

If indeed it turns out there is no difference in cell lysis (or not), the authors should test pooled supernatant+pellet blot to test whether processed cytokines are not being released. As shown currently (ELISA and westerns on supernatants), the authors haven't ruled out the possibility that there may be processed intracellular IL-1b/IL-18.

We have now addressed these concerns by using a later timepoint to assess IL-1 β and caspase-1 release, again showing that P22077 blocks both their processing and release. We were unable to detect any cleaved forms of either protein in the lysates and thus only the pro- forms have been shown. We have included the full blots for the supernatants (updated Fig 3C).

6. The BRET assay is quite interesting and should be developed further. However, I have several questions about the assay which could be addressed by additional controls. For example, I would like to see that their assays show defects when

using ASC mutants that do not oligomerise or bind to NLRP3. My major concern is that the authors use HEK293 cells for these experiments, and these cells are known to not express NLRP3. How is Nigericin leading to activation in this system? Reconstitution of HEK cells with various inflammasome proteins for mutational studies has been done before and this is still a valid biochemical assay system. However, the authors should note that nigericin causes golgi collapse and other toxicity issues even in the absence of NLRP3. In order to show specificity of their BRET2 assay, therefore, they must carry out assays with co-transfected NLRP3 (and ideally inactive mutants lacking the PYD region) to establish that their assay is a faithful reporter of signal-driven ASC assembly.

We agree with the reviewer that the BRET assay needs to be developed further. The use of HEK cells to model inflammasome activation has been previously used, however we are aware that HEK cells and macrophages are very different. We also agree with this reviewer on the use of ASC mutants and we would have also liked to use ubiquitin single mutants to better describe the type of ubiquitin chains here involved. On reflection, we have decided to remove the BRET data from the manuscript. Both reviewer 2 and reviewer 3 have raised different concerns about this data that we could not address within the actual time frame. We therefore felt it was best to try to focus on assessing the ubiquitination of ASC in a more relevant cell context, hence the work with the THP-1 cells expressing ASC-mRFP (which is now shown in Fig. 4E).

8. In continuation with point 6, the authors should show changes to ubiquitylation of endogenous ASC in macrophages due to USP inhibitor treatments so as to correlate findings from the BRET assay. The BRET2 assay in HEK293 is not a sufficient measure of what might be going on in MDMs.

We thank the reviewer for their suggestion. As we have mentioned with the previous reviewers comment, we agree that an assessment of ASC in the endogenous setting was important to the paper and to that end, we used THP-1 cells expressing ASC-mRFP and utilised the RFP-TRAP system for purification. This enabled us to blot for Ub. By using this system we have been able to show that ASC, in a macrophage cell line, is deubiquitinated upon nigericin treatment that is reversed in the presence of P22077. This data is now part of a revised figure 4 (Fig. 4E), and importantly confirmed the results obtained in the BRET experiments.

9. The ABP approach, which is again quite interesting, would seem to suggest that the compounds also inhibit other USPs, although the authors discuss that the irreversibility of their probes. In such a scenario, it would be good to try nigericin treatment in excess K⁺ to block activation and show that the probes can reveal specific activation with nigericin ie. the K⁺ efflux required for NLRP3 activation somehow activates USPs?

We are happy to read that the reviewer found this data interesting, and we apologise for any confusion in regards to the activity-based probes. In general, our description of how the probes function was not as clear as we thought, and we have re-written the text to make this clearer (Lines 227-241). To address this reviewer's concerns a better description of what the probes are actually doing might help. The ABPs are non-specific and bind to the active site of many different USPs as well as other DUB family members (de Jong et al., 2012). Our comments regarding the irreversibility of the probes reflect our experiments where we use the probes to show that P22077 can inhibit the activity of both USP7 and USP47. In these experiments, the inhibitor and the chemical probes are essentially competing to bind to the catalytic site of the two DUBs. As the VME- group on the chemical probes is known to irreversibly bind to these proteins, whereas P22077 is known to only bind reversibly, the probe can effectively outcompete P22077, which means that we are likely under-estimating the inhibition of activity that P22077 imparts in figure 5C. Several other publications have used these probes to assess the DUB targets of P22077 (Altun et al., 2011; Ritorto et al., 2014) and provide evidence to suggest

P22077 at higher concentrations can inhibit other DUBs. This is why it was important for us to use CRISPR/Cas9 to validate our findings.

We thank the reviewer for their suggestion to try and see whether using high extracellular K⁺ with nigericin would inhibit the activity increases we observe. We have now performed these experiments and they now form figure 5D. Intriguingly even though the excess K⁺ was sufficient to block the inflammasome activation, as shown by IL-1 β release, we observed no changes in the activity of USP7 and USP47. This suggests that K⁺ efflux does not control the activity changes in the two DUBs and another mechanism must be involved. Although this is interesting, any further work to understand the mechanisms controlling such activity changes would be beyond the scope of this publication, but could be an exciting avenue for future work.

Have the authors tested that the bands correspond to USP7/47 by trying similar assays in the gRNA-based KO cells?

We apologise if we did not make this clear in the manuscript. If we carry out these experiments in KO cells, there are no additional bands upon the addition of the probes given that there is no USP7 or USP47 to bind to. The higher molecular weight bands correspond to DUB that is bound to the probe, and there are three pieces of evidence present in the paper to support this. Firstly, in all conditions tested, the higher molecular weight bands are not present where no probe has been added (see the panels named 'Unlabelled' in figure 5). Secondly, addition of P22077 increases the levels of the lower molecular weight band (or the unbound DUB), demonstrating that these bands are specific to these DUBs. Thirdly, the bands representing the lower molecular weight in these figures are the same size as those that are decreased upon sgRNA induction shown in all of our figures (Figs. 6, EV6 and EV7). Additionally, these probes have been extensively used in publications showing both molecular weight shifts in Western blots and DUB specificity in MS analysis (Altun et al., 2011; de Jong et al., 2012)

10. The authors should at least additionally test AIM2 inflammasomes in the CRISPR KO cells and show that endogenous ASC ubiquitylation is altered in these cells.

We agree with the reviewer that this experiment was important to the paper and thank them for their suggestion. We have thus now performed Aim2 activating experiments in KO cells (now Fig. EV7). We were unable to achieve a sufficient IP of endogenous ASC in THP-1 cells and have included an assessment of ASC ubiquitination through ASC-mRFP expressing cells. Even though we have not performed these experiments in CRISPR/Cas9-edited cells, we have still been able to provide evidence that USP7 and USP47 may contribute to ASC deubiquitination in a macrophage setting.

11. Statistics: the authors must clarify that when they mean n=10 for ELISAs etc (for example in Fig legends), these are 10 means from biologically independent experiments and do not include technical replicates. Only means from independent experiments should be used for statistical calculations.

The reviewer makes an important point here and we apologise for not making this clearer from the onset. We have edited the manuscript now so that all of the figure legends describe exactly what the n numbers define. Note, where MDMs were used, the n number represents individual blood donors. Where BMDMs were used, the n number represents individual murine donors.

Minor points.

1. pg 3 - middle para - authors mean large number of DUBs (rather than high number)

We have now changed the text in the manuscript and use the word

'numerous' instead.

2. pg 4 line 10 from top (line numbers would have been useful throughout) - in this instance and other places where the authors discuss sequence relatedness or similarities, they use the term 'homology'. They must refrain from using this incorrect term even if widely used. Homology means related by descent - a sequence descended from an ancestral gene or not (gene is a daughter of a parent or not) - there is no such thing as partial homology. However, there can be partial sequence similarities which can be expressed quantitatively simply by stating % sequence similarity. They should state % protein sequence similarity between the USPs throughout and not use the word homology.

Firstly we agree that line numbers would have been useful and have now included these.

We thank the reviewer for flagging up our incorrect use of the term homology and have corrected this in the text (lines 67-70; Fig EV1).

3. pg 9 - line 4 - different instead of deferent.

This has now been corrected.

Referee #3:

This paper investigates two deubiquitinase enzymes in the regulation of NLRP3 inflammasome responses, and shows that USP7 and USP47 are both activated in response to NLRP3 triggering. They are implicated in the response by use inhibitors but importantly also CRISPR knockout (Figure 6) and an activity assay confirms activation of the enzymes under inflammasome triggering conditions (Figure 5). Figure 4 is an attempt to show ASC as a target of the deubiquitinases, but it fails to convince. The authors have attempted a BRET technique with overexpression of components in HEK cells. It is not clear why they have done this, without including any data on endogenous ASC ubiquitination in macrophages. Figure 4 is the major problem with the paper, as panel B gives no confidence in any of the data of this figure. (Please note that although I would rate the technical quality of most of the paper as high, I would rate this figure as unacceptable). Inspection of reference 57 on which they base this technique gives clear saturation curves that are at a distinctly higher signal than the negative control that increases linearly. The data presented here show random noise not significantly above the negative control, and curve fitting to this data is not realistic. The authors themselves do not seem confident enough of this data to include it in the abstract. Otherwise, most of the paper is well executed, but at this stage should conclude that these deubiquitinases promote NLRP3 inflammasome activity, through an unknown target.

Specific Comments

Figure 4 should be omitted. If you have any data from macrophages to show effects on endogenous ubiquitination, that would be much superior, and would raise the level of the paper substantially. In the absence of that, the target of the deubiquitinases remains not established.

We agree that more needs to be done to optimize this BRET approach. Given that both reviewer 2 and reviewer 3 have raised different concerns about this data that could not address within the actual time frame we have decided to remove the BRET data from the manuscript. We felt it was best to try to focus on assessing the ubiquitination of ASC in a more relevant cell context, hence the work with the THP-1 cells expressing ASC-mRFP (which is now shown in Fig. 4E).

Figure 2. Panel B is not what the legend says. It shows nigericin treatment apparently, and not LPS treatment. If it was +/- LPS treatment, we would see pro-IL-1b induced. I am not sure what this is showing.

We thank the reviewer for picking this up. It actually was an error and referred to LPS treated cells +/- P22077. The figure and figure legend have now been amended and now also includes +/- LPS treatment +/- P22077 as suggested by referee 2.

Figure 3. There is no guarantee within this figure that the chemically transfected polydA-dT (that is giving a very low response) is actually recognized by AIM2, and this is only presumed by extension from mouse work. However, this response has not been well explored in human cells. This would not be so important except that effects on both NLRP3 and AIM2 pathways are used as a justification that the DUBs are acting on something affecting ASC, and not something further upstream in the NLRP3 pathway.

As we have previously mentioned for the first reviewer's comment we have now addressed the issue of not detecting cell death in AIM2 inflammasome-activated human cells. We believed that our issue was likely that of sensitivity and that by using a longer activation times with dAdT, we were able to detect LDH release, as well as higher levels of IL-1 β release that were blocked by P22077. We agree that it was important to address these concerns, as it is a justification to hypothesise that ASC is our likely DUB target. To that end we added an additional set of experiments in our USP7/47 deficient THP-1 cells to supplement these findings. We also extended this to look at the response in murine BMDMs where we activated both the AIM2 and NLRC4 inflammasome, with protocols that are well established (as referred to by the reviewer here). We saw that P22077 blocked both of these inflammasomes, in both WT BMDMs and in those that were NLRP3 KOs, which we believe further supports our original hypothesis.

Some legends say n=11 individual experiments. Others just say "n=11" and the level at which replication is done is not clear. Please clarify in every legend. For example, n=8 donors or n=8 experiments etc.

As with reviewer 2, this reviewer makes an important point here and we apologise for not making this clearer from the onset. We have edited the manuscript now so that all of the figure legends describe exactly what the n numbers define.

If the authors are convinced that ASC ubiquitination is important, then this question is relevant. If DUBs were to regulate the polymerization of ASC, then this infers that a substantial proportion of ASC should be ubiquitinated, and this should be seen just on a western blot for ASC. Are there higher bands on total ASC blots that are consistent with ubiquitination, and that are altered following inflammasome triggering? Or otherwise, if ASC ubiquitination is only of a small minority of the protein, how would this have an important effect on the behavior of the bulk ASC?

We agree with the reviewer that this experiment was both important and relevant to the paper and thank them for their suggestion. We have thus now performed experiments to try and study ASC ubiquitination (in a relevant macrophage cell line; Fig. 4E). We tried overexposing total ASC blots in the different conditions we test in the paper, and failed to visualise smears that are indicative of poly-ubiquitination. We were able to detect such smears when we employed an IP of ASC-mRFP in THP-1 cells that overexpress this fusion protein. Even with this setup the intensity of the ubiquitin smears is still very low, and we took many steps to ensure we could achieve the maximal sensitivity (Emmerich and Cohen, 2015). Thus it is a fair and valid question to raise the idea of the stoichiometry of ASC ubiquitination and how relevant that is functionally. That said, the difficulty of detecting ASC ubiquitination by Western blotting might not be directly related to the level of ubiquitination that truly exists on the protein and there are many factors that can affect this. There is always the question that the Ub antibody is not as sensitive as the ASC antibody, for example. Another issue is that due to the nature of the smearing you see that result from the different lengths of polyubiquitination of ASC, the signal of the protein is essentially diluted across a broad area of the membrane, which greatly reduces the

signal to noise ratio. Thus while we believe it will be important to know how much of the ASC is ubiquitinated/deubiquitinated, we do not have the resources to answer that. Ultimately, the key question is (as the reviewer refers to) the biological importance of the event. We would argue that as we see a functional effect of both P22077 and USP7/USP47 deficiency, if ASC is indeed the target of these DUBs, then their action is important enough to result in a disruption of their action.

Figure 3C and D. What is the "ASC input"? The Figure legend does not fully describe what has been done here, as the methods state that you are actually looking at the ASC speck pellets isolated by centrifugation. In Figure 3C, this should not say lysates, but should say ASC speck pellets isolated by 2400g centrifugation. So at what stage is the "ASC input" taken? It is obviously not the ASC speck fraction without crosslinking, as ASC is in all samples in panel C. Is it the lysate after 300g spin? With the lysis conditions used (isotonic, and 21G needle), I expect pyroptotic cells would readily lyse, but the control cells would not. Did you check that cells actually lysed with these conditions? Also in Fig 3D, why are there two panels for ASC crosslinked- are they different exposures?

We thank the reviewer to bringing this point of confusion to our attention and are sorry this was not clear. We largely followed the protocol described in (Fernandes-Alnemri et al., 2007) so that the 'input' represents the protein lysate in an isotonic solution, physically sheared with a 21G needle, as stated by the reviewer. We are confident that this provides a sufficient lysis of the cells, as no cells could be seen when we looked at this solution under a microscope. Additionally, we observed a similar sized pellet (that represents the nuclear – insoluble fraction) in each of the different conditions, and we see equal amounts of ASC by WB using this buffer. To make this clearer, the figure legend (now Fig 4A and B) has now been modified to reflect this:

“Western blots showing ASC-oligomerisation after DSS-mediated crosslinking in insoluble fractions from cell lysates of LPS-primed (1 µg/mL, 4 hrs.) MDMs preincubated with either 0.1% DMSO or P22077 (2.5 µM) 15 mins. before treatment with nigericin (10 µM, 45 mins.). The 'input' represents the abundance of ASC in the soluble fraction, prior to isolation of the insoluble fraction.”

And we have also modified the supernatant legend to make this clearer...

“Western blots showing ASC-oligomerisation after DSS-mediated crosslinking in supernatants from MDMs treated as in D. Blots show membranes that were probed with an anti-ASC antibody. The two boxes shown for the crosslinked samples represent two different exposure times. The input represents non-crosslinked supernatant.”

Minor points

Figure 3F. How you got error bars and stats here is not clear, and should be explained in the legend, not just in the methods. Provide n and an explanation please.

We agree with the reviewer and have now provided this in each figure legend.

Discussion - "requires nucleate on-induced polymerization". Is this supposed to be "nucleation"?

Yes, this has now been amended.

Methods- cells and treatments - problem with exponent on cell concentration; also "plaquets"- do you mean platelets?

Yes, these have also been amended.

Although I think this data should be taken out, note that filters for luciferase and GFP are listed at the same wavelength. Also "the milliBRET units are calculated as the percentage of BRET signal relative to basal signal, and results are expressed as the average of mBU after nigericine treatment". What is meant by the "basal signal"? How can anything called a "milli-unit" be a percentage? The processing of data for

this is not at all easy to understand.
We have now removed this data from the paper.

Figure 1H legend "PS-primed" typo
We thank the reviewer for noticing this and have corrected this.

References

- Altun, M., H.B. Kramer, L.I. Willems, J.L. McDermott, C.A. Leach, S.J. Goldenberg, K.G. Kumar, R. Konietzny, R. Fischer, E. Kogan, M.M. Mackeen, J. McGouran, S.V. Khoronenkova, J.L. Parsons, G.L. Dianov, B. Nicholson, and B.M. Kessler. 2011. Activity-based chemical proteomics accelerates inhibitor development for deubiquitylating enzymes. *Chem Biol* 18:1401-1412.
- Baroja-Mazo, A., F. Martin-Sanchez, A.I. Gomez, C.M. Martinez, J. Amores-Iniesta, V. Compan, M. Barbera-Cremades, J. Yague, E. Ruiz-Ortiz, J. Anton, S. Bujan, I. Couillin, D. Brough, J.I. Arostegui, and P. Pelegrin. 2014. The NLRP3 inflammasome is released as a particulate danger signal that amplifies the inflammatory response. *Nat Immunol* 15:738-748.
- Compan, V., F. Martin-Sanchez, A. Baroja-Mazo, G. Lopez-Castejon, A.I. Gomez, A. Verkhratsky, D. Brough, and P. Pelegrin. 2015. Apoptosis-associated specklike protein containing a CARD forms specks but does not activate caspase-1 in the absence of NLRP3 during macrophage swelling. *J Immunol* 194:1261-1273.
- de Jong, A., R. Merckx, I. Berlin, B. Rodenko, R.H. Wijdeven, D. El Atmioui, Z. Yalcin, C.N. Robson, J.J. Neefjes, and H. Ovaa. 2012. Ubiquitin-based probes prepared by total synthesis to profile the activity of deubiquitinating enzymes. *Chembiochem* 13:2251-2258.
- Emmerich, C.H., and P. Cohen. 2015. Optimising methods for the preservation, capture and identification of ubiquitin chains and ubiquitylated proteins by immunoblotting. *Biochem Biophys Res Commun* 466:1-14.
- Evavold, C.L., J. Ruan, Y. Tan, S. Xia, H. Wu, and J.C. Kagan. 2017. The Pore-Forming Protein Gasdermin D Regulates Interleukin-1 Secretion from Living Macrophages. *Immunity*
- Fernandes-Alnemri, T., J. Wu, J.W. Yu, P. Datta, B. Miller, W. Jankowski, S. Rosenberg, J. Zhang, and E.S. Alnemri. 2007. The pyroptosome: a supramolecular assembly of ASC dimers mediating inflammatory cell death via caspase-1 activation. *Cell Death Differ* 14:1590-1604.
- Fink, S.L., and B.T. Cookson. 2006. Caspase-1-dependent pore formation during pyroptosis leads to osmotic lysis of infected host macrophages. *Cellular microbiology* 8:1812-1825.
- Franklin, B.S., L. Bossaller, D. De Nardo, J.M. Ratter, A. Stutz, G. Engels, C. Brenker, M. Nordhoff, S.R. Mirandola, A. Al-Amoudi, M.S. Mangan, S. Zimmer, B.G. Monks, M. Fricke, R.E. Schmidt, T. Espevik, B. Jones, A.G. Jarnicki, P.M. Hansbro, P. Busto, A. Marshak-Rothstein, S. Hornemann, A. Aguzzi, W. Kastanmuller, and E. Latz. 2014. The adaptor ASC has extracellular and 'prionoid' activities that propagate inflammation. *Nat Immunol* 15:727-737.
- Fuller, S.J., L. Stokes, K.K. Skarratt, B.J. Gu, and J.S. Wiley. 2009. Genetics of the P2X7 receptor and human disease. *Purinergic signalling* 5:257-262.
- Ritorto, M.S., R. Ewan, A.B. Perez-Oliva, A. Knebel, S.J. Buhrlage, M. Wightman, S.M. Kelly, N.T. Wood, S. Virdee, N.S. Gray, N.A. Morrice, D.R. Alessi, and M. Trost. 2014. Screening of DUB activity and specificity by MALDI-TOF mass spectrometry. *Nat Commun* 5:4763.

Thank you for the submission of your revised manuscript to EMBO reports.
 We have meanwhile received a complete set of reviews from all referees, which I include below for

your information.

As you will see, the referees acknowledge that the revised manuscript has been considerably strengthened. However, I also note that two major concerns remain after the revision:

-) The IP for endogenous ASC failed and the referees are not fully convinced that ASC is a direct target of USP7 and USP47.
-) The referees note that a recent paper showed that DNA transfection fails to activate AIM2 in human monocytes.

I have further discussed these points with the referees and they all agree that it will be important to provide further evidence that USP7/47 deubiquitinate endogenous ASC. Referee 1 suggested to use a mouse specific antibody and referee 2 made some other suggestions in his/her report (see below). Moreover, during this discussion the referees emphasized that it will be important to test whether DNA transfections indeed activate the AIM2 or rather the NLRP3 pathway. This could be done using MCC950 on MDMs transfected with poly dA:dT. Referee 3 further suggested to look at AIM2 in THP1 cells. The recent paper by the Hornung lab (PMID 29033128) apparently showed that DNA transfection activates AIM2 in this cell type. If this response is not inhibited by MCC950 but by USP knockout or inhibitor treatment, it would provide further evidence that USP7/47 regulate both, NLRP3 and AIM2. Please note that Figure 3 should not be removed from the manuscript.

Please also comment on the different concentration of P22077 that is necessary to affect mouse AIM2 and NLRC4 responses (2.5 and 10 μ M, Fig EV2 and Fig EV4) since the referees were also concerned that different concentrations are required even though the response is thought to be mediated by ASC in both cases.

In summary, given the overall positive evaluation of your manuscript, I would like to give you the opportunity to address the remaining concerns as outlined above and in the referee reports. Please address all referee concerns in a complete point-by-point response. Revised manuscripts should be submitted within three months of a request for revision; they will otherwise be treated as new submissions. Please contact us if a 3-months time frame is not sufficient for the revisions so that we can discuss the revisions further.

Moreover, browsing through the manuscript myself, I noticed several things that need to be resolved upon resubmission:

- Please provide the manuscript text as a Word file.
- Please note that only 5 figures can be displayed in the Expanded View modus. Please choose 5 figures that you want to promote to Expanded View and provide all additional supplementary material in a single pdf labeled Appendix. The Appendix includes a table of content with page numbers on the first page, all figures and their legends. Please follow the nomenclature Appendix Figure Sx for these figures throughout the text and also label the figures according to this nomenclature. For more details please refer to our guide to authors.
- Figure 4D: please indicate the number of samples analysed. If x cells were counted from 1 experiment, please note that the number of independent experiments is 1 in this case and a statistical evaluation can thus not be applied. It is advisable to display the actual data points in such cases in e.g. a scatter blot. Moreover, I noticed a mistake in the figure legend. You refer to "...images depicted in F." I assume that the correct reference is "C".
- Thank you for submitting source data. This is highly appreciated and the files will be published along with your manuscript. I however noticed some inconsistencies, as follows:
 - +) Source data for Fig. 3C: the labeling for IL-1b seems wrong. The label in the SD file says " IL-1b lysate" while the corresponding panel is displayed as "supernatant" in the figure.
 - +) SD for Fig. 5C: the labels for USP7 (unlabeled, labeled) do not match with the corresponding figure.
 - +) You supplied source data for Figure EV4B which does not display Western blot data.
- Please indicate the source of the human peripheral blood mononuclear cells in the Methods section. Were these obtained by you or did you purchase them?

- Since you used wildtype and NLRP3-mutant mice, please fill in the relevant sections in the Author Checklist (D-Animal models).

- Please provide a running title (max. 40 characters including spaces)

Please let me know if you have any further questions. I am looking forward to receiving a revised version of your manuscript.

REFEREE REPORTS

Referee #1:

In the rebuttal letter, the author wrote that they could not IP endogenous ASC in THP-1 using 4 different commercial antibodies. In J774 cells or BMDMs, did they try the mouse specific anti-ASC (D2W8U) to IP ASC, then detection of ubiquitin chains with the FK2 antibody? I guess this anti-ASC works in IP in mouse cells. The P22077 can be used to confirm the involvement of USP7/47.

Very recently, an interesting paper has described that in human myeloid cells, AIM2 is dispensable for DNA-mediated inflammasome activation. Instead, detection of cytosolic DNA by the cGAS-STING axis induces a cell death program initiating potassium efflux upstream of NLRP3 (Gaidt et al. Cell 2017, PMID 29033128). In figure 3, to confirm that the transfection of poly dA:dT does activate AIM2 but not the NLRP3 inflammasome in hMDMs, MCC950 should be used. As both AIM2 or NLRP3 require ASC for inflammasome formation, the use of P22077 is efficient in preventing caspase-1 and IL-1 β processing.

Referee #2:

This is the revised version of their manuscript describing the roles of USP7 and USP47 in inflammasome activation. The authors have improved the manuscript by providing several western blots that convincingly show the effect of these USPs in the activation of NLRP3 and AIM2 inflammasomes. Removal of the BRET2 assay has not significantly impacted the paper, and is for the better if additional controls couldn't be included. The authors include an assay showing ubiquitylation of Asc-mRFP and its inhibition by the compound as these experiments with endogenous ASC did not work despite using several commercial antibodies. However, there remain inconsistencies and lack of mechanisms which should be addressed. Main points.

1. The authors repeatedly claim USP7 and USP47 do not affect transcriptional priming of inflammasomes. While overall this is shown using the long and short LPS treatments, they should clearly state the effects of the compound on the transcription of pro-IL1 β and pro-IL18 etc. The authors appear to use long and indirect sentences to overlook this even though I do not think it affects the novelty of their findings.
2. In the data shown as fold-change of mRNA, the graph in Fig. 2A should be plotted to clearly show the lower part around 1. The data and blot suggest slightly reduced pro-IL18 as well but the authors do not comment on this. In the associated Fig. 2B, NLRP3 is almost completely induced with LPS, but the fold-change relative to untreated in the RT-PCR is none - this is inconsistent with reported upregulation of NLRP3 by LPS and their own western blots. The lack of an effect of MCC950 on pyroptosis is surprising (see PMID: 27521339).
3. Similarly, the authors see a 2-fold increase in USP7 message, and a similar trend, but insignificant by statistics, on USP47 in Fig. EV5B. Yet they mention this is vague terms and point to the lack of difference in THP1 cells. I would suggest saying 2-fold rather than "very modest" on line 222 and elsewhere with similar data to strictly present facts than their qualitative interpretations. As above, any transcriptional effects do not compromise their findings.
4. It is a pity that endogenous ASC IPs did not work. In the absence of this data, the authors should provide additional data to support their central USP target. The ubiquitylation is shown in one western blot which isn't convincing. The rest of the study is of high technical quality, and I would have expected an RNAi approach to convincingly show that the increased ubiquitylation can be

abolished when USP7 and USP47 are silenced. The authors should mention that reduced ASC oligomerisation could be simply due to reduced activation and pyroptosis and it doesn't provide further mechanistic insight from ASC oligomerisation in lysates (line 198).

5. Or provide some more insight on the proposed link between USPs and ASC - for example, the authors say USP7 cleaves K11/48/63 chains - they could check if any of these increased on the IP of Asc-mRFP. Is the inflammasome activation inhibited similarly by P22077 in the Asc-mRFP stable cell line? If ASC is the target, I would expect the efficacy of the inhibitors to be reduced in the cell line that overexpressed Asc - is this the case?

Minor points.

1. I think the term USP7/USP47 shouldn't be used as this kind of "/" is usually indicative of two names of the same gene. I suggest spelling out USP7 and USP47 throughout.
2. Legend to Fig 4 has subpanel alphabets errors that should be corrected.
3. The sentence on line 297 should be rephrased "non-NLRP3 inhibitory effect of P22077" - they mean to say they wanted to test another inflammasome in the CRISPR lines.

Referee #3:

The authors have obviously worked hard on this and addressed many of the reviewers' questions quite adequately. However, Figure 4E stands out as a real problem. This is the only piece of data used to justify the idea that these DUBs affect ASC ubiquitination directly. Apart from a big background problem, there is no clear result to be gained here. There also seems to be a bit of a blank spot that is not so evident on the file shown in the additional figure data file. I am not sure how the contrast could have been adjusted to obtain the final figure. This bears no resemblance to the ASC ubiquitination data in reference 21. Overall, this blot does nothing to convince me that ASC is a target of these enzymes, or indeed that ASC is polyubiquitinated. I think the rest of the data may stand, but the picture of how these DUBs affect inflammasome function is still unclear. Where I have stated that figure quality is unacceptable, this is the figure to which I refer.

Other comments:

1. Line 145 "IL-1b mRNA levelswas reduced by about half with P22077". This is hard to read with the cut axis, but levels actually look to be reduced from about 1150 to 200. I make that reduced by about 80%. Please be more precise than "about half". The authors state that it hasn't made much difference to protein levels, but a number of figures show some difference, that could have some impact on interpretation of the level of cleaved IL-1b.

2. Figure 2 raw western blot images - NLRP3 and proIL1b are swapped around compared to the actual figure 2. Which is correct?

3. Unfortunately for Figure 3, Veit Hornung's group has just published the lack of AIM2 inflammasome response in monocytes and primary human bone marrow macrophages, and the fact that this response to transfected DNA is mediated via activation of NLRP3 and is inhibited by MCC950 (Cell 2017 171:1-15). They didn't show MDM, but MDM express almost undetectable AIM2, less than monocytes. Consequently Figure 3 should be omitted as a proof of anything related to AIM2.

4. There are two pages labelled EV4 raw western blot images, and I think the first one is EV2. The raw images have overlaid light images of the markers, but then a lot of reflection and flaws in the image. Maybe better to put the actual images that were used without the markers?

5. Line 182-183. The response should not be stated as NLRC4-dependent as this has not been formally shown here. Also, you should not say that the inhibitor did not inhibit cell death, rather there was no detectable cell death above background.

6. Line 208. "as previously described" requires a reference

7. Figure EV7. It is not clear why procaspase-1 expression is varying so much with LPS and DNA treatment. What is regulating this?

8. It would help if figures using THP1 noted that they are PMA differentiated.
9. Line 374 "...lead to eventual destabilisation of the plasma membrane". Membrane destabilisation is caused by Gasdermin D downstream of caspase-1. This sentence is not well written.
10. How would you envisage your work fitting with reference 21?
11. In reply to reviewer 1, the authors say that USP inhibition affects IL-1 β processing but not AIM2 cell death. It is more likely that under your conditions you just have cell death that is not specific for AIM2 because you are looking at long time points when many other changes including lipofectamine toxicity may be playing a role. If you maintain that NLRP3-dependent cell death is inhibited then any AIM2-dependent death would also have to be- it is just all dependent on gasdermin D that is activated similarly in both cases.
12. The authors in rebuttal say that they could not get an MDM response to flagellin - I think that is expected, as an acknowledged difference between mouse and human NLRC4?
13. Figure 4A figure legend. I think that you should say "The "input" representsin the TOTAL LYSATE prior to isolation of the insoluble fraction", not the "soluble fraction".
14. I feel the discussion is a bit long and repetitive of results.

2nd Revision - authors' response

24 May 2018

Referee #1:

In the rebuttal letter, the author wrote that they could not IP endogenous ASC in THP-1 using 4 different commercial antibodies. In J774 cells or BMDMs, did they try the mouse specific anti-ASC (D2W8U) to IP ASC, then detection of ubiquitin chains with the FK2 antibody? I guess this anti-ASC works in IP in mouse cells. The P22077 can be used to confirm the involvement of USP7/47.

We agree that this would have been a good approach and something we could do in the future. However based on the results described above where we could not confirm the involvement of human AIM2 in USP7 and/or USP47 mediated inflammasome activation we did not pursue this any further. Instead we focused on determining the effect of P22077 on NLRP3 oligomerization and ubiquitination which is now shown in Fig. 4.

Very recently, an interesting paper has described that in human myeloid cells, AIM2 is dispensable for DNA-mediated inflammasome activation. Instead, detection of cytosolic DNA by the cGAS-STING axis induces a cell death program initiating potassium efflux upstream of NLRP3 (Gaidt et al. Cell 2017, PMID 29033128). In figure 3, to confirm that the transfection of poly dA:dT does activate AIM2 but not the NLRP3 inflammasome in hMDMs, MCC950 should be used. As both AIM2 or NLRP3 require ASC for inflammasome formation, the use of P22077 is efficient in preventing caspase-1 and IL-1 β processing.

We thank this reviewer for pointing this out. We have now performed these experiments and found that activation by Poly (dA:dT) transfection in MDMs and in our THP1s is due to NLRP3 given that MCC950 inhibits its activation and this inhibition is not potentiated by P22077 (Fig EV3).

Referee #2:

This is the revised version of their manuscript describing the roles of USP7 and USP47 in inflammasome activation. The authors have improved the manuscript by providing several western blots that convincingly show the effect of these USPs in the activation of NLRP3 and AIM2 inflammasomes. Removal of the BRET2 assay has not significantly impacted the paper, and is for the better if additional controls couldn't be included. The authors include an assay showing ubiquitylation of Asc-mRFP and its inhibition by the compound as these experiments with endogenous ASC did not work despite using several commercial antibodies.

However, there remain inconsistencies and lack of mechanisms which should be addressed.

Main points:

1. The authors repeatedly claim USP7 and USP47 do not affect transcriptional priming of inflammasomes. While overall this is shown using the long and short LPS treatments, they should clearly state the effects of the compound on the transcription of pro-IL1b and pro-IL18 etc. The authors appear to use long and indirect sentences to overlook this even though I do not think it affects the novelty of their findings.

We agreed that the text needed rewording and we have now addressed this comment.

2. In the data shown as fold-change of mRNA, the graph in Fig. 2A should be plotted to clearly show the lower part around 1. The data and blot suggest slightly reduced pro-IL18 as well but the authors do not comment on this.

This has now been addressed and we have plotted the graph to make this clearer. The levels of IL-18 showed more variability than other genes. In some donors LPS seemed to induce a clear increase in IL-18 levels, but this was not observed in others. This is probably due to the donor to donor variation we observe in humans. We observed a similar thing with proteins levels, but overall observed no clear reduction with P22077. This has now been included in the text. Page6, Line 185.

In the associated Fig. 2B, NLRP3 is almost completely induced with LPS, but the fold-change relative to untreated in the RT-PCR is none - this is inconsistent with reported upregulation of NLRP3 by LPS and their own western blots.

Similarly to IL-18 we observed variability in human donors and this big difference in NLRP3 expression levels were not always observed. We also found in a RNA sequencing study (comparing untreated to PMA/LPS treated THP1 cells, not published) that RNA levels of NLRP3 were only increased by 1.7 fold, similar to the data obtained by qPCR presented in the manuscript. Most of the NLRP3 molecular mechanisms described so far have been studied in mouse. As we have just observed, with the example of human and mouse AIM2, it is possible that human and mouse NLRP3 regulation is also different. More experiments would be needed in order to confirm this, but is beyond the scope of the work hereby presented.

The lack of an effect of MCC950 on pyroptosis is surprising (see PMID: 27521339).

We agree with the reviewer here. In fact a reduction on ATP-induced cell death when MCC950 is present is observed (Fig EV1 C) but it is not statistically significant. We have now changed the text to ... "A reduction on ATP-induced cell death was observed with the well characterized NLRP3 inhibitor MCC950 although this was not statistically significant." Page 4, line 124.

3. Similarly, the authors see a 2-fold increase in USP7 message, and a similar trend, but insignificant by statistics, on USP47 in Fig. EV5B. Yet they mention this is vague terms and point to the lack of difference in THP1 cells. I would suggest saying 2-fold rather than "very modest" on line 222 and elsewhere with similar data to strictly present facts than their qualitative interpretations. As above, any transcriptional effects do not compromise their findings.

We agree with the reviewer, and apologise for the non-objective language. These changes have now been made, page 8, line 258.

4. It is a pity that endogenous ASC IPs did not work. In the absence of this data, the authors should provide additional data to support their central USP target. The ubiquitylation is shown in one western blot which isn't convincing. The rest of the study is of high technical quality, and I would have expected an RNAi approach to convincingly show that the increased ubiquitylation can be abolished when USP7 and USP47 are silenced.

As we have mentioned above, and in the light of the new results using MCC950 in Poly (dA:dT) inflammasome activation we have now focused on the effect of USP7 and USP47 on NLRP3. We have now included new data on NLRP3. We observed that the oligomerization of NLRP3 after ATP treatment is affected by USP7 and USP47 inhibition and that this inhibition affects NLRP3-ubiquitination status. We have performed these experiments in murine BMDMs given that the pull down of NLRP3 was more efficient and detection of its ubiquitination was much more sensitive in these cells than in THP1. Additionally, we had experimental issues detecting NLRP3 oligomerization in THP1 cells after nigericin treatment (unfortunately these cells do not respond to ATP), and thus cannot provide this data for our KO cells.

The authors should mention that reduced ASC oligomerisation could be simply due to reduced activation and pyroptosis and it doesn't provide further mechanistic insight from ASC oligomerisation in lysates (line 198).

We agree that this needed to be mentioned to avoid confusion and has now been included in Page 7, line 221.

5. Or provide some more insight on the proposed link between USPs and ASC - for example, the authors say USP7 cleaves K11/48/63 chains - they could check if any of these increased on the IP of Asc-mRFP.

We have not done that for ASC, but instead we have tested the effect of P22077 on NLRP3-K63-ubiquitination (Fig. 4), showing that there is a small increase in abundance of this modified form upon inhibition.

Is the inflammasome activation inhibited similarly by P22077 in the Asc-mRFP stable cell line? If ASC is the target, I would expect the efficacy of the inhibitors to be reduced in the cell line that overexpressed Asc - is this the case?

Although we have not included this data in the new version of the paper, we did measure the effect of P22077 on this cell line for the ubiquitination experiments. We observed reduced inhibition in cell death with 2.5uM compared to what we observe in WT cells, confirming the suggestion of the reviewer. Yet these cells are clonally derived, and we have observed different rates of both activation and inhibition with clonally derived cells with other lab projects, and thus we would be reluctant to make any clear conclusions based on this data.

Minor points:

1. I think the term USP7/USP47 shouldn't be used as this kind of "/" is usually indicative of two names of the same gene. I suggest spelling out USP7 and USP47 throughout. We would like to thank the reviewer for this suggestion. This has now been changed throughout.

2. Legend to Fig 4 has subpanel alphabets errors that should be corrected. This has now been corrected. Former Fig. 4 has now changed to Fig. 3.

3. The sentence on line 297 should be rephrased "non-NLRP3 inhibitory effect of P22077" - they mean to say they wanted to test another inflammasome in the CRISPR lines. This has now been changed and amended according to new results.

Referee #3:

The authors have obviously worked hard on this and addressed many of the reviewers' questions quite adequately. However, Figure 4E stands out as a real problem. This is the only piece of data used to justify the idea that these DUBs affect ASC ubiquitination directly. Apart from a big background problem, there is no clear result to be gained here. There also seems to be a bit of a blank spot that is not so evident on the file shown in the additional figure data file. I am not sure how the contrast could have been adjusted to obtain the final figure. This bears no resemblance to the ASC ubiquitination data in reference 21. Overall, this blot does nothing to convince me that

ASC is a target of these enzymes, or indeed that ASC is polyubiquitinated. I think the rest of the data may stand, but the picture of how these DUBs affect inflammasome function is still unclear. Where I have stated that figure quality is unacceptable, this is the figure to which I refer.

ASC-ubiquitination has been shown in other papers (besides the USP50 one):

- DOI: 10.1084/jem.20132486 , <http://jem.rupress.org/content/211/7/1333>, Fig 6.

- doi:10.4049/jimmunol.1402851, <http://www.jimmunol.org/content/jimmunol/194/10/4880.full.pdf>, Fig 3A
- doi: 10.1038/ni.2215, <https://www.ncbi.nlm.nih.gov/pmc/articles/PMC4116819/>, Fig 6.

In all of these figures, ASC ubiquitination is described as a smear over 85kDa (which is also observed in the USP50 paper (fig 3D)). We observed an accumulation of the ubiquitination smear over 250kDa. Unlike in all of these publications, here we pulled down ASC-RFP which size is bigger than endogenous ASC so this could explain the differences seen here and in this other papers.

However, we agree that this data was unconvincing and in light of our results pointing to a role for NLRP3 activation after comments from this and other reviewers, we have removed this data.

Other comments:

1. Line 145 "*IL-1 β mRNA levelswas reduced by about half with P22077*". This is hard to read with the cut axis, but levels actually look to be reduced from about 1150 to 200. I make that reduced by about 80%. Please be more precise than "about half". The authors state that it hasn't made much difference to protein levels, but a number of figures show some difference, that could have some impact on interpretation of the level of cleaved IL-1 β .

We agree with the reviewer, and again apologise for using non-objective language. The figure (Fig 2A) has now been changed so this can be clearly seen. Indeed we observed that there is a reduction on IL-1 β mRNA levels and as the reviewer said this could have some impact on the level of cleaved IL-1 β . That is why we do two things throughout the paper, first we prime cells with LPS in the absence of USP7 and USP47 inhibitors, then remove the media and add the appropriate inhibitor for 15 min before adding the inflammasome activator. In this way we remove any potential effect of the inhibitor on IL-1 β transcription. Secondly we measure caspase-1 processing by western blot. This is independent of the levels of IL-1 β and a very good indicator of inflammasome activation. This is especially relevant when using USP7 and USP47 deficient cells where we cannot avoid the potential effect of USP7 and USP47 on IL-1 β expression.

2. Figure 2 raw western blot images - NLRP3 and proIL1b are swapped around compared to the actual figure 2. Which is correct?

We would like to thank the reviewer for spotting this, as we had missed it. The files have now been amended and the right versions uploaded.

3. Unfortunately for Figure 3, Veit Hornung's group has just published the lack of AIM2 inflammasome response in monocytes and primary human bone marrow macrophages, and the fact that this response to transfected DNA is mediated via activation of NLRP3 and is inhibited by MCC950 (*Cell* 2017 171:1-15). They didn't show MDM, but MDM express almost undetectable AIM2, less than monocytes. Consequently Figure 3 should be omitted as a proof of anything related to AIM2.

This reviewer is right and we have now amended the paper accordingly, as discussed above. Figure 3 has now been changed to Fig. EV3, where we now show that MCC950 blocks Poly (dA:dT) induced IL-1 β release, and hence the effect we observe after activation with this stimuli is due to the activation of the cGAS-STING-NLRP3 axis instead of AIM2.

4. There are two pages labelled EV4 raw western blot images, and I think the first one is EV2. The raw images have overlaid light images of the markers, but then a lot of reflection and flaws in the image. Maybe better to put the actual images that were used without the markers?

We would like to thank the reviewer for picking this up. This was a mistake and has now been amended. Regarding the reflection and flaws in the image we have now included the image without the markers so it is clearer what the actual image looked like, as suggested by the reviewer.

5. Line 182-183. The response should not be stated as NLRC4-dependent as this has not been formally shown here. Also, you should not say that the inhibitor did not inhibit cell death, rather there was no detectable cell death above background.

This has now been amended. Page 5, Line 159.

6. Line 208. "as previously described" requires a reference.

We would like to thank the reviewer for the suggestion, however this is now not relevant as we have removed this part of the text.

7. Figure EV7. It is not clear why procaspase-1 expression is varying so much with LPS and DNA treatment. What is regulating this?

I believe this could be due to interferons (INF) regulating caspase-1 levels although we have not done the experiments to confirm this. dsDNA can activate NF- κ B as well as IRF3 and IRF7 when sensed by TLRs or cytosolic receptor such as the cGAS-STING pathway (which we now know is activated by human macrophages in response to Poly (dA:dT)) producing interferons (PMID:

19362700). We have observed in other experiments in the lab that $\text{INF}\gamma$ increases the levels of caspase-1 in non-hematopoietic cells. Also, when checking caspase-1 (Casp1) mRNA levels using ImmGen (Immunological Genetic Project: <http://www.immgen.org/>) it shows that INF up-regulates caspase-1 levels in several immune cell types (see figure below), suggesting this could be the case.

8. It would help if figures using THP1 noted that they are PMA differentiated. This has now been included.

9. Line 374 "...lead to eventual destabilisation of the plasma membrane". Membrane destabilisation is caused by Gasdermin D downstream of caspase-1. This sentence is not well written. We agree with this and it has now been re-written. This has now been changed to "...all of which result in caspase-1 activation and the creation of membrane pores by gasdermin-D eventually leading to pyroptosis [51]"; page 12, line 408.

10. How would you envisage your work fitting with reference 21?

Although this is now less relevant given our latest results implicating NLRP3, the fact that different USPs (or other deubiquitinases) can regulate ubiquitination of ASC would not be surprising. The targets and function of many DUBs are still unknown and it would be possible that different DUBs recognise the same target. It is also possible that although they recognise the same target they may edit ubiquitin chains in a different manner or have selectivity to different ubiquitin chains adding different levels of posttranslational regulation to ASC and consequently the inflammasome.

11. In reply to reviewer 1, the authors say that USP inhibition affects $\text{IL-1}\beta$ processing but not AIM2 cell death. It is more likely that under your conditions you just have cell death that is not specific for AIM2 because you are looking at long time points when many other changes including lipofectamine toxicity may be playing a role. If you maintain that NLRP3-dependent cell death is inhibited then any AIM2-dependent death would also have to be- it is just all dependent on gasdermin D that is activated similarly in both cases.

The reviewer was right. We now know that cell death was not AIM2 specific and was mediated by the cGAS-STING pathway (PMID 29033128), as discussed above. We now also know that our THP1 cells also responded to poly (dA:dT) through this pathway since $\text{IL-1}\beta$ release was blocked by MCC950 in these cells. Therefore this cGAS-STING-mediated lysosomal cell death is upstream of NLRP3 and hence would explain why we observe no inhibition in cell death but inhibition of inflammasome activation in response to Poly (dA:dT). This is similar to our results with CPPD crystals (Fig. 1), where cell death (that is inflammasome independent) is not blocked by P22077 but inflammasome activation is. In the case of murine BMDMs we observed minimal induction of cell death by Poly (dA:dT) and this was not significantly altered by P22077 (although there was a slight decrease overall). Although these are very different cells, it is also important to remember that

treatment time points are also different with 4 hrs Poly (dA:dT) stimulation in BMDMs compared to the 24 hrs stimulation we used in THP1 cells.

12. The authors in rebuttal say that they could not get an MDM response to flagellin - I think that is expected, as an acknowledged difference between mouse and human NLRC4?

The reviewer is right and this has been published before (PMID: 23940371). We apologise for not having acknowledged this before.

13. Figure 4A figure legend. I think that you should say "The "input" representsin the TOTAL LYSATE prior to isolation of the insoluble fraction", not the "soluble fraction".

We are sorry that this was not clear. What we mean is that input refers to the cell lysate before centrifugation to isolate ASC-specks in the insoluble fraction. We have now changed this to: "The 'input' represents the abundance of ASC in cell lysates prior to isolation of the insoluble fraction."

14. I feel the discussion is a bit long and repetitive of results.

We have now changed the discussion to address this and incorporate our new results and make it less repetitive from results.

3rd Editorial Decision

25 June 2018

Thank you for the submission of your further revised manuscript to EMBO reports. We have meanwhile received a complete set of reviews from all referees, which I include below for your information.

As you will see, the referees acknowledge that the revised manuscript has been considerably strengthened and that the current dataset looks more convincing. However, the newly added data also raises further concerns as outlined by referee 3. Again, I discussed the referee reports with all three referees and both, referee 1 and 2 agree with the concerns raised by the third referee and think that all concerns are pertinent and should be addressed. Some of the concerns can be addressed by clarification and text changes. I would like to emphasize though that the control IP (referee 3, point 4) is absolutely essential, both referee 3 and 2 emphasized this again in their further feedback.

In summary, given the overall positive evaluation of your manuscript, I would like to give you the exceptional opportunity to address these remaining concerns as outlined above and in the referee reports in a final round of revisions. Please address all referee concerns in a complete point-by-point response. Please indicate all changes to text and figures in the point-by-point response and please also use track changes in the manuscript text (or indicate the changes using colored or highlighted text). I would also like to point out that as per editorial policy, EMBO reports will reassess novelty if revisions are received more than six months from the date when the initial decision letter was sent.

Revised manuscripts should be submitted within three months of a request for revision; they will otherwise be treated as new submissions. Please contact us if a 3-months time frame is not sufficient for the revisions so that we can discuss the revisions further.

I look forward to seeing a revised version of your manuscript when it is ready. Please let me know if you have questions or comments regarding the revision.

REFEREE REPORTS

Referee #1:

I guess that this revised version is now suitable for publication in EMBO reports

Referee #2:

This is a second revision of this manuscript and the authors appear to have addressed most major criticisms that were raised. The new experiments appear to point towards the regulation by UPSs of NLRP3 rather than ASC which was suggested in previous versions. The new data on NLRP3

ubiquitylation and the impact of P22077 on this is a lot more convincing than previous data on ASC ubiquitylation. While the K63 blot is not the best, the increase in K63 chains with higher concentrations of the compound is visible in the higher part of the blot. I only have minor comments on this version which is acceptable for publication.

minor points

1. line 398, comma should be after however (...however,...)
2. line 234, comma needed after As expected.
3. The authors should be careful when they say NLRP3 oligomers in the text. Perhaps 'oligomeric complex' or 'higher molecular weight complex containing NLRP3' is more accurate because they do not know what else is contributing to the higher Mw complex on native gels (in addition to hyper-ubiquitylation of NLRP3 which contributes to high Mw)

Referee #3:

Once again the paper is improved. However, some more problems are introduced during modifications. Specifically:

1. Effect of inhibitor on the non-canonical inflammasome. Figure EV4: The text says "P22077 did not affect the ability of cytosolic LPS to trigger IL-1 β release and pyroptosis, unlike MCC950 that blocked these two events". Transfected LPS induces caspase-11-mediated pyroptosis (but caspase 11 does not cleave IL-1b), and during this process the K⁺ efflux apparently activates NLRP3, which then mediates IL-1b cleavage. The expected result would be that MCC950 and P22077 would both inhibit IL-1b release that is downstream of NLRP3, but not the cell death that is purely dependent on caspase-11.

There are several problems here:

- (i) I presume that there is an error in the figure, and that the last group of 3 bars is supposed to have a "+" symbol for LPS as well as lipofectamine and MCC950? Statements below assume this.
- (ii) The text is wrong regarding the effect on cell death shown in the figure. MCC950 did not inhibit the modestly induced cell death and it is not expected to.
- (iii) I would have expected MCC950 and P22077 should both inhibit IL-1b release that is NLRP3-dependent downstream of the non-canonical inflammasome. P22077 does not. MCC950 does (assuming that the labelling on the figure is wrong). I think this needs drawing out a bit more as an unexpected finding, as current opinion is that non-canonical and most canonical NLRP3 activation critically involve K⁺ efflux, and the underlying mechanism is expected to be the same. Please note in the results if something is unexpected, or else it makes reading it quite confusing. However, the fact that line 169 says "in this experiment we observed that ...this was the only time this was observed throughout the study." suggests that Fig EV4 might be the result of a single experiment. If this is the case, this requires repetition before publication.

2. Fig 2A was modified. The actual numbers plotted on the graph for IL-18 and IL1b are now quite different from what they were in the second revision, as are the error bars. Since this was only supposed to be re-plotting the same data, this needs explanation. For some reason, there is now less effect of the inhibitor on IL-1b levels. Also, the two statistical comparisons shown seem unlikely to have the same level of significance.

3. Figure 4A. This result is clearly the opposite of what was anticipated, in that you hoped the inhibitor would prevent formation of clustered NLRP3. It would help if you clearly acknowledge in the results that this went against the hypothesis, and does not actually assist the argument, but is valid data.

4. Figure 4B. There is no control to allow determination of how much of the ubiquitination signal is associated with NLRP3. Despite pre-clearing, immunoprecipitation still generally brings down lots of proteins. The best control would be the same experiment in parallel on NLRP3 knockout BMM. However, either an irrelevant antibody control or omission of NLRP3 antibody would help make this assessment.

5. Figure 4B. Doesn't the K63-linked Ub result look more hopeful than the poly-Ub, since it is having an effect at 2.5 μ M, where you are seeing inhibition of the NLRP3 response? Also, both

signals decrease with ATP - why might this be? Lastly, the BRCC3 paper has gone much further in linking BRCC3 directly with NLRP3, and they didn't pick up USP7 in their screen. Perhaps USP7 and 47 regulate BRCC3 rather than NLRP3 directly?

Minor points

Line 333 please reference the Hornung paper again when you refer to the cGAS-STING mediated pathway to remind the reader. Also in line 339.

Line 182 please put "mRNA" rather than "transcriptional"

3rd Revision - authors' response

3 July 2018

Referee #1:

I guess that this revised version is now suitable for publication in EMBO reports

We would like to thank the reviewer for considering that the manuscript is suitable for publication.

Referee #2:

This is a second revision of this manuscript and the authors appear to have addressed most major criticisms that were raised. The new experiments appear to point towards the regulation by UPSs of NLRP3 rather than ASC which was suggested in previous versions. The new data on NLRP3 ubiquitylation and the impact of P22077 on this is a lot more convincing that previous data on ASC ubiquitylation. While the K63 blot is not the best, the increase in K63 chains with higher concentrations of the compound is visible in the higher part of the blot. I only have minor comments on this version which is acceptable for publication.

We thank the reviewer for considering the paper is acceptable for publication.

Minor points

1. line 398, comma should be after however (...however,...)

This has now been included.

2. line 234, comma needed after As expected.

This has now been included.

3. The authors should be careful when they say NLRP3 oligomers in the text. Perhaps 'oligomeric complex' or 'higher molecular weight complex containing NLRP3' is more accurate because they do not know what else is contributing to the higher Mw complex on native gels (in addition to hyper-ubiquitylation of NLRP3 which contributes to high Mw)

We agree with the reviewer that we don't know what else can be contributing to the higher MW complexes and have now changed the term NLRP3 oligomer to oligomeric complexes containing NLRP3 in line 391 and 395; page 12. Similarly we have also included the term higher molecular weight complexes containing NLRP3 in line 240.

Referee #3:

Once again the paper is improved. However, some more problems are introduced during modifications. Specifically:

1. *Effect of inhibitor on the non-canonical inflammasome. Figure EV4: The text says "P22077 did not affect the ability of cytosolic LPS to trigger IL-1 β release and pyroptosis, unlike MCC950 that blocked these two events". Transfected LPS induces caspase-11-mediated pyroptosis (but caspase 11 does not cleave IL-1b), and during this process the K+ efflux apparently activates NLRP3, which then mediates IL-1b cleavage. The expected result would be that MCC950 and P22077 would both inhibit IL-1b release that is downstream of NLRP3, but not the cell death that is purely dependent on caspase-11.*

There are several problems here:

(i) *I presume that there is an error in the figure, and that the last group of 3 bars is supposed to have a "+" symbol for LPS as well as lipofectamine and MCC950? Statements below assume this.*

We apologise for this error in the figure. The last group of 3 bars should have a '+' indicating the presence of LPS, and we have now amended this.

(ii) The text is wrong regarding the effect on cell death shown in the figure. MCC950 did not inhibit the modestly induced cell death and it is not expected to.

We again apologise for this. The referee is correct and as expected no inhibition of cell death is observed with MCC950. This has now been amended and changed to "...P22077 did not affect the ability of cytosolic LPS to trigger IL-1 β release, unlike MCC950 that blocked this event while neither of these inhibitors affected the levels of cell death (Fig EV4).", line 168, page 5.

(iii) I would have expected MCC950 and P22077 should both inhibit IL-1 β release that is NLRP3-dependent downstream of the non-canonical inflammasome. P22077 does not. MCC950 does (assuming that the labelling on the figure is wrong). I think this needs drawing out a bit more as an unexpected finding, as current opinion is that non-canonical and most canonical NLRP3 activation critically involve K⁺ efflux, and the underlying mechanism is expected to be the same. Please note in the results if something is unexpected, or else it makes reading it quite confusing.

We agree with the reviewer that this is an unexpected result and it really surprised us. As we have included in the discussion (now line 362, page 11) a recent paper described very similar results where the absence of the E3 ubiquitin ligase Pellino2 affects canonical NLRP3 activation but not non-canonical activation (Supplementary Fig4; <https://doi.org/10.1038/s41467-018-03669-z>). Our results, together with this paper, suggest that there are more differences between canonical and non-canonical NLRP3 activation than previously appreciated, and this will need further study. As suggested by the referee we have now included the following in the result section, line 170, page 5:

"This effect on IL-1 β was an unexpected result given that current knowledge states that the underlying mechanisms regulating non-canonical and canonical NLRP3 activation are expected to be the same and depend on K⁺ efflux."

However, the fact that line 169 says "in this experiment we observed that ...this was the only time this was observed throughout the study." suggests that Fig EV4 might be the result of a single experiment. If this is the case, this requires repetition before publication.

Sorry for the confusion. This is not a single experiment, when we say "this experiment" we are referring to the non-canonical activation experiment. Non-canonical activation experiments in the presence of these inhibitors was performed in BMDMs derived from 3 independent murine donors ($n=3$ as stated in the EV4 figure legend). We have now changed it to make it clear line 172, page 5:

"We observed that 10 μ M P22077 induced an increase in basal cell death compared to 2.5 μ M or vehicle treated cells. This increase in cell death induced by 10 μ M P22077 was not observed in other experiments (FigEV3) and was likely due to a drug independent effect."

2. Fig 2A was modified. The actual numbers plotted on the graph for IL-18 and IL1 β are now quite different from what they were in the second revision, as are the error bars. Since this was only supposed to be re-plotting the same data, this needs explanation. For some reason, there is now less effect of the inhibitor on IL-1 β levels. Also, the two statistical comparisons shown seem unlikely to have the same level of significance.

The referee is right in that the numbers are now different and I apologise for not having explained this before. When we went to reformat the graph for the third revision I realised that the method used to calculate the fold change increase was not what we had used in the first submission ($\Delta\Delta$ Ct method as stated in the methods section). I had to then recalculate the fold changes to fit this method and replot the data as it stands now. I would like to reassure this referee that the RAW data is the same as previous submission and the overall results do not change.

I apologise for not having clarified this in the text. The statistical analysis performed is one-way ANOVA analysis (using Prism/Graphpad) comparing the mean of each column to the LPS treated sample, as we were interested to see the effect of P22077 on LPS. This gave an equal level of significance when comparing LPS vs P22077 than LPS vs LPS/P22077. We have now included this in Fig2 legend "...using a one-way ANOVA analysis comparing the mean of each column to the LPS treated sample" (line 761, page 23).

3. Figure 4A. This result is clearly the opposite of what was anticipated, in that you hoped the inhibitor would prevent formation of clustered NLRP3. It would help if you clearly acknowledge in the results that this went against the hypothesis, and does not actually assist the argument, but is valid data.

I agree that the results from this experiment might be the opposite of what one would expect if we assume that to prevent ASC-speck formation we need to prevent the formation of NLRP3-clusters. However although we expected fewer oligomers of NLRP3, we could also expect more Ub-NLRP3,

which could be attributed to the higher bands in the native gel blots. To address this unexpected result we have now included the following sentence in the text (page 241, page7):

“This was surprising given that, based on the current literature (39, 40) and the fact that P22077 prevented ASC-speck formation, one would assume that USP7 and USP47 inhibition would prevent formation of NLRP3-containing oligomers. However, as the increase...”

4. Figure 4B. There is no control to allow determination of how much of the ubiquitination signal is associated with NLRP3. Despite pre-clearing, immunoprecipitation still generally brings down lots of proteins. The best control would be the same experiment in parallel on NLRP3 knockout BMM. However, either an irrelevant antibody control or omission of NLRP3 antibody would help make this assessment.

As mentioned above, we agree with the referee that the IP control experiment was necessary. In fact, we had already run this control in our experiments but we decided to exclude it from the final figure for clarity. As a control we used an IgG2b antibody that matched the IgG type from the anti-NLRP3 (instead of the anti-NLRP3) used for immunoprecipitation. We used this in cell lysates that had been pre-treated with P22077 at 10 μ M since this is the condition where we observed the greatest increase in ubiquitination. In this control pulldown we observed some background ubiquitination when using the anti-Poly-Ub antibody suggesting there might be other proteins binding to the IgG or beads as the reviewer 3 suggested, but this is still much lower than the ubiquitination observed when using the anti-NLRP3 antibody. When we used the anti-K63-Ub antibody minimal background ubiquitination was observed. We have now included this control in Fig4B. We have also described this in the results section (Line 250, Page 8), fig4B legend (page 24, line 814) and methods section (Line 482 and 600).

5. Figure 4B. Doesn't the K63-linked Ub result look more hopeful than the poly-Ub, since it is having an effect at 2.5 μ M, where you are seeing inhibition of the NLRP3 response? Also, both signals decrease with ATP - why might this be?

Although initially the K63 looks like a more promising target, we don't believe that this is the case. Our data suggests that there might be constitutive K63-deubiquitination of NLRP3 by USP7 and USP47 and we know from our activity based data (Fig5) that USP7 and USP47 can be active at basal levels explaining why we see increased ubiquitination of NLRP3 in resting LPS-primed cells treated with P22077. However we observed that after ATP treatment this ubiquitination increase disappears even in the presence of P22077. We believe this could be due to the effect of other DUBs such as BRCC3 (<https://doi.org/10.1016/j.molcel.2012.11.009>) which would not be blocked by P22077 and still be able to deubiquitinate the receptor. We still don't understand how different DUBs contribute to NLRP3 activation and the impact that impairing the activity of one would have on others. In fact in this BRCC3 paper by Py et al. it was shown that BRCC3 knockdown increased ubiquitination of NLRP3 in LPS-primed cells (similar to what we observe with our inhibitor) however how the ubiquitination status on NLRP3 changed after inflammasome activation was not shown and hence we don't know if they observed a similar effect on K63-linked chains. The fact that we did not observe any changes in NLRP3 protein levels in the presence of P22077 suggests that K48-linked ubiquitin chains are not involved either. In addition to this, USP7 and USP47 can also cleave K11-linked chains, something which we have not specifically tested here. So it is also possible that these DUBs target this type of chain, following activation, and contribute to the P22077-mediated increase in NLRP3 ubiquitination detected when using the Pan-Ub antibody. We believe that at this stage we cannot conclude which type of chain is definitively involved in this process as further and deeper studies are needed to address this. We have now changed the results (line 252, Page 8) and the discussion (Line 402, page 12) to acknowledge this.

Lastly, the BRCC3 paper has gone much further in linking BRCC3 directly with NLRP3, and they didn't pick up USP7 in their screen. Perhaps USP7 and 47 regulate BRCC3 rather than NLRP3 directly?

In the BRCC3 paper they perform a screen by testing co-immunoprecipitation of NLRP3 with different DUBs in HEK cells. Only the DUBs that showed interaction with NLRP3 were moved forward and its ability to affect NLRP3 ubiquitination studied. As the reviewer said, USP7 was not picked up in this screen. However not all DUB-target interactions will be stable enough to be detected by co-IP as some DUB interactions are known to be transient or weak affinity protein-protein interactions. Hence the ability of any DUB that has this sort of interaction with NLRP3 will not have been studied. In addition, USP47 was not tested at all. With our current data, we cannot exclude the possibility that USP7 and/or USP47 are acting on BRCC3 or any other target. Our data shows that USP7 and USP47 regulate the NLRP3 inflammasome but we cannot confirm whether this is a direct or indirect effect.

Minor points

Line 333 please reference the Hornung paper again when you refer to the cGAS-STING mediated pathway to remind the reader. Also in line 339.

This has now been included, now lines 346 and 351.

Line 182 please put "mRNA" rather than "transcriptional"

This has now been changed, now line 185.

4th Editorial Decision

25 July 2018

Thank you for the submission of your further revised manuscript to EMBO reports. It has been sent back to former referee 3 who is now also supportive of publication. Please clarify if the results shown in Figure EV4 are from 3 independent experiments or if the cells from the 3 donors were isolated on the same day and treated in parallel in one experiment. If the latter is true then the number of independent experiments is actually one and the experiment should be repeated.

From the editorial side, there are also a few things that we need before we can proceed with the official acceptance of your paper.

REFEREE REPORT

Referee #3:

- Figure EV4. If this is 3 independent experiments, that is fine. If it is one experiment with cells from 3 mice, that is not fine.

- I think the ability to process data of Figure 2A in two different ways and get graphs that look quite different is a cautionary tale. Can I suggest for future work, using just delta Ct method (not delta delta) and showing data only normalised to the control gene (GAPDH etc). Presentation of data with less processing is more informative.

4th Revision - authors' response

2 August 2018

Please clarify if the results shown in Figure EV4 are from 3 independent experiments or if the cells from the 3 donors were isolated on the same day and treated in parallel in one experiment. If the latter is true then the number of independent experiments is actually one and the experiment should be repeated.

Results shown in Figure EV4 were obtained after treating cells from different donors in the same day. We have now repeated this again in 2 different days using 2 different donors each time, so we have now 3 independent experiments and the new figure has been included and the text amended (line 175). The main message is still the same; P22077 does not block non-canonical inflammasome.

Corresponding Author Name: Gloria Lopez-Castejón

Manuscript Number: EMBOR-2017-44766V1